# Sexual rejection via a vomeronasal receptor-triggered limbic circuit

Takuya Osakada[1,2], Kentaro K. Ishii[1,2], Hiromi Mori[1,2], Ryo Eguchi[1,2], David M. Ferrero[3], Yoshihiro Yoshihara[2,4], Stephen D. Liberles[3], Kazunari Miyamichi[1,2,6] & Kazushige Touhara[1,2,5]

Mating drive is balanced by a need to safeguard resources for offspring, yet the neural basis for negative regulation of mating remains poorly understood. In rodents, pheromones critically regulate sexual behavior. Here, we observe suppression of adult female sexual behavior in mice by exocrine gland-secreting peptide 22 (ESP22), a lacrimal protein from juvenile mice. ESP22 activates a dedicated vomeronasal receptor, V2Rp4, and V2Rp4 knockout eliminates ESP22 effects on sexual behavior. Genetic tracing of ESP22-responsive neural circuits reveals a critical limbic system connection that inhibits reproductive behavior. Furthermore, V2Rp4 counteracts a highly related vomeronasal receptor, V2Rp5, that detects the male sex pheromone ESP1. Interestingly, V2Rp4 and V2Rp5 are encoded by adjacent genes, yet couple to distinct circuits and mediate opposing effects on female sexual behavior. Collectively, our study reveals molecular and neural mechanisms underlying pheromone-mediated sexual rejection, and more generally, how inputs are routed through olfactory circuits to evoke specific behaviors.

[1] Department of Applied Biological Chemistry, Graduate School of Agricultural and Life Sciences, The University of Tokyo, Tokyo 113-8657, Japan. [2] ERATO Touhara Chemosensory Signal Project, JST, The University of Tokyo, Tokyo 113-8657, Japan. [3] Department of Cell Biology, Harvard Medical School, Boston 02115 Massachusetts, USA. [4] RIKEN Center for Brain Science, Saitama 351-0198, Japan. [5] International Research Center for Neurointelligence (WPI-IRCN), The University of Tokyo Institutes for Advanced Study, Tokyo 113-0033, Japan. [6]Present address: RIKEN Center for Biosystems Dynamics Research, Hyogo 650-0047, Japan. These authors contributed equally: Takuya Osakada, Kentaro K. Ishii, Hiromi Mori, Kazunari Miyamichi. Correspondence and requests for materials should be addressed to K.M. (email: kazunari.miyamichi@riken.jp) or to K.T. (email: ktouhara@mail.ecc.u-tokyo.ac.jp)

Reproduction can bring heavy burdens associated with pregnancy, nursing, and rearing of offspring. Females across the animal kingdom may choose not to mate in order to preserve energy and resources for more desirable partners and/or to increase maternal investment in the care of existing offspring[1,2]. Sexual behavior can be influenced by various factors, such as hunger, stress, and also chemical signals emitted from other conspecifics in the environment[2–4]. Particularly in rodents, olfactory signals, such as pheromones, are thought to play an important role in regulating reproduction[3,4]. Decades of studies in rodents have identified pheromones that positively influence female sexual behavior. These include volatile and proteinaceous compounds in male urine that attract female mice[5–8] and a male-specific lacrimal protein called exocrine gland-secreting peptide 1 (ESP1)[9], which enhances sexually receptive posturing (termed lordosis) in female mice[10]. In contrast to the positive signals, relatively little is known about the pheromones, or even their sources, that negatively affect female sexual behavior.

Classic studies using lesioning and electrical stimulation of brain areas in rats have identified not only facilitatory, but also inhibitory systems, that control female sexual receptivity. A well-characterized facilitatory center of lordosis is located in the ventromedial hypothalamus (VMH), in particular, in the steroid hormone receptor-expressing neurons located in the ventrolateral area of the VMH (VMHvl). Lesioning of VMHvl or cell type-specific ablation of estrogen or progesterone receptor-expressing neurons in VMHvl completely abolished lordosis[11–13], whereas electrical stimulation of VMHvl facilitated lordosis[14]. In contrast, lordosis-inhibiting systems are suggested to be located in the medial preoptic area (MPA) and lateral septum, and lesioning of these areas enhanced the lordosis response[15]. Although a recent study made progress in characterizing the neural activities of VMHvl neurons in female mice[16], it is largely unclear what external stimuli drive lordosis-inhibiting systems to negatively influence female reproduction.

The vomeronasal organ (VNO) plays a critical role in mediating chemosensory signals that influence sexual behaviors[17]. There are 187 and 121 G-protein coupled receptors that belong to type 1 (V1R) or type 2 (V2R) vomeronasal receptors, respectively, in the mouse genome[18,19]. Ligands for the vast majority of vomeronasal receptors are unknown, limiting our understanding of sensory mechanisms underlying behavioral specificity. VNO signals are conveyed to the brain, first to the accessory olfactory bulb (AOB), and then to the limbic system nuclei, such as the amygdala, that coordinate behavioral responses[20]. How a specific behavioral output is elicited by a given vomeronasal input is mostly unknown. Our studies of ESP1 have begun to elucidate the neural basis by which a pheromone enhances female sexual behavior. ESP1 is detected by a single type of V2R, V2Rp5 (also known as Vmn2r116)[10], in the VNO. ESP1-induced enhancement of lordosis is then mediated by a labeled-line neural circuit that includes the medial amygdala posteroventral part (MeApv), ventromedial hypothalamus dorsal part (VMHd), and the dorsal periaqueductal gray (dPAG) regions in the midbrain[10,13]. This circuit seems to work in parallel with VMHvl and adjusts the level of sexual receptivity of female mice.

In this study, we aimed to identify a pheromone signal that negatively influences sexual behavior of female mice, as well as the receptor basis and neural circuitry by which the pheromone modulates choice of behaviors. As a strong candidate, we analyzed a juvenile proteinaceous pheromone called ESP22, which we recently reported to negatively affect sexual behavior of male mice[21]. ESP22 is released into tears of 2 to 3-weeks old mice and acts as an immaturity signal to prevent unwanted mounting by adult male mice. Although ESP22 can also be received by adult female mice, responses in females remained elusive. Here, we show that ESP22 induces a prolonged suppression of sexual receptivity in adult female mice. We also identify a specific receptor of ESP22 and a limbic neural circuit responsible for negatively regulating sexual behavior in female mice.

## Results

**ESP22 suppresses sexual receptivity of virgin female mice**. We investigated the effects of ESP22 on the sexual behavior of virgin female mice. Adult C57BL/6J virgin female mice in a hormone-primed pseudo-estrus state (see Methods section) were pre-exposed to cotton swabs with 50 μg of ESP22 or control buffer (Fig. 1a). We started sexual behavior assays 30 min after the cotton exposure to stabilize female sexual behavior, according to other pheromone-mediated behavioral tests[10,22,23]. ESP22-exposed female mice showed significantly fewer lordosis responses to mounting episodes by a stud ICR male mouse (Fig. 1b, c). Intriguingly, we observed that an ESP22-stimulated female mouse frequently showed at least one of the following postures: standing, crouching down, keeping their limb tight, or turning their body to inhibit insertion when male mice tried to mount onto female mice (Supplementary video 1). We defined this behavior as rejection. The number of rejections and the ratio of rejections normalized to the number of male mountings (rejection ratio) were significantly higher in the ESP22-exposed females (Fig. 1b, c). As a result, there was a longer latency to male insertion and significant decrease in the number of male intromissions in this group (Fig. 1c). These results indicate that ESP22 suppresses the sexual behavior of virgin female mice. There was no effect on the proportion and duration of the estrus cycle in female mice (Supplementary Fig. 1a, b), and we did not observe any apparent changes in social interaction[24] after exposure to ESP22 (Supplementary Fig. 1c, d), excluding the indirect effects of ESP22.

We previously determined differences in ESP22 production among mouse strains; C57BL/6 juveniles produced and secreted ESP22, whereas C3H/HeJ mice did not[21]. We then investigated whether sexual rejection behavior was evoked by a physiological amount of ESP22. Virgin females who were pre-exposed to C57BL/6J juvenile mice that secreted ESP22[21]. exhibited a significant reduction in lordosis responses and a trend of increased rejection responses compared to those pre-exposed to C3H/HeJ juvenile mice who did not secrete ESP22 (Fig. 1d–f). We also performed sexual behavior assays in C57BL/6J lactating mothers and found that they exhibited rejection behaviors in most mounting attempts by male mice (Supplementary Fig. 1e–g). This suggests that one of the targets of ESP22 secreted by 2 to 3-week-old juvenile mice is non-breeding female mice in their environment. Collectively, these data demonstrate that ESP22 derived from juvenile mice can negatively regulate the sexual receptivity of adult female mice, which leads to increased unsuccessful mating.

**ESP22 counteracts ESP1-mediated sexual facilitation**. Both our previous and current studies have demonstrated that ESP1 and ESP22 exhibit opposing effects to female sexual behaviors (Fig. 1)[10]. In nature, female mice can be exposed to both pheromones in temporal proximity. To mimic this situation experimentally, we serially exposed females to ESP1 followed by ESP22 (Fig. 2a). The exposure did not result in an enhancement of the lordosis response (Fig. 2b, c), which demonstrated that ESP1-dependent lordosis enhancement was fully suppressed by ESP22. We additionally analyzed the effects of ESP22 on breeding efficiency through continuous application of ESP22 in the drinking water and using ESP1 as a control (Fig. 2d). The parturition day

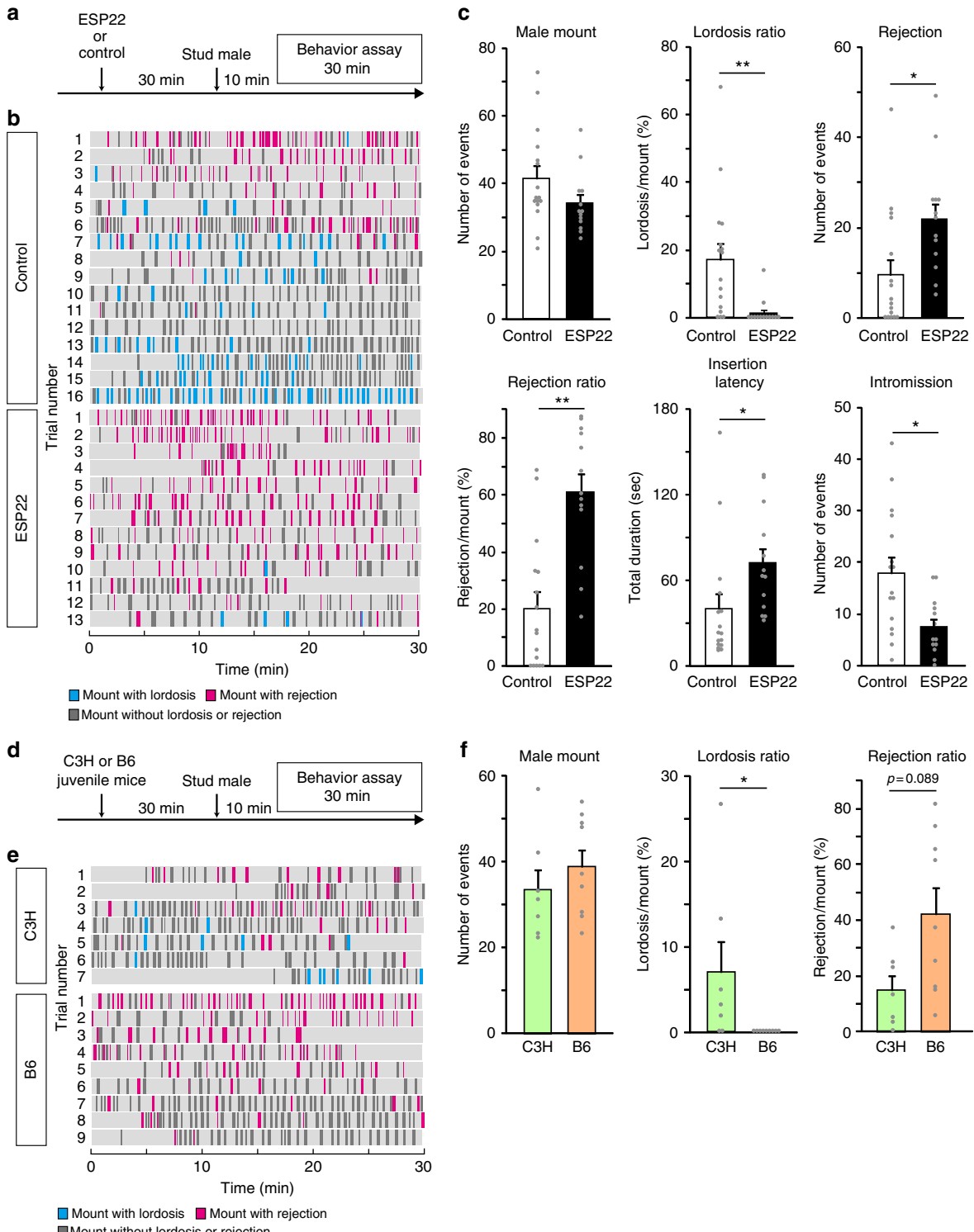

**Fig. 1** ESP22 suppresses sexual receptivity of virgin female mice. **a** Timeline for sexual behavior assays using adult female mice pre-exposed to either ESP22 or control buffer. **b** Raster plot representing mounting episodes made by the male mouse, with cyan and magenta bars representing attempts associated with lordosis or rejection responses, respectively, by C57BL/6J female mice. Mounting attempts without lordosis or rejection are represented with grey bars. Control buffer-exposed female mice, $n = 16$; ESP22-exposed female mice, $n = 13$. **c** Quantification of the sexual behaviors of control buffer- and ESP22-exposed female mice. Each dot represents data of an individual female mouse. **d** Timeline for the sexual behavior assays using C57BL/6J adult female mice pre-exposed to either C3H/HeJ (C3H, ESP22-) or C57BL/6 (B6, ESP22+) juvenile mice. **e**, **f** Raster plots and quantification are the same as detailed in **b**, **c**. C3H juvenile-exposed female mice, $n = 7$; B6 juvenile-exposed female mice, $n = 9$. Error bars, S.E.M. $^{**}p < 0.01$ and $^*p < 0.05$ by Wilcoxon rank sum test with Holm correction

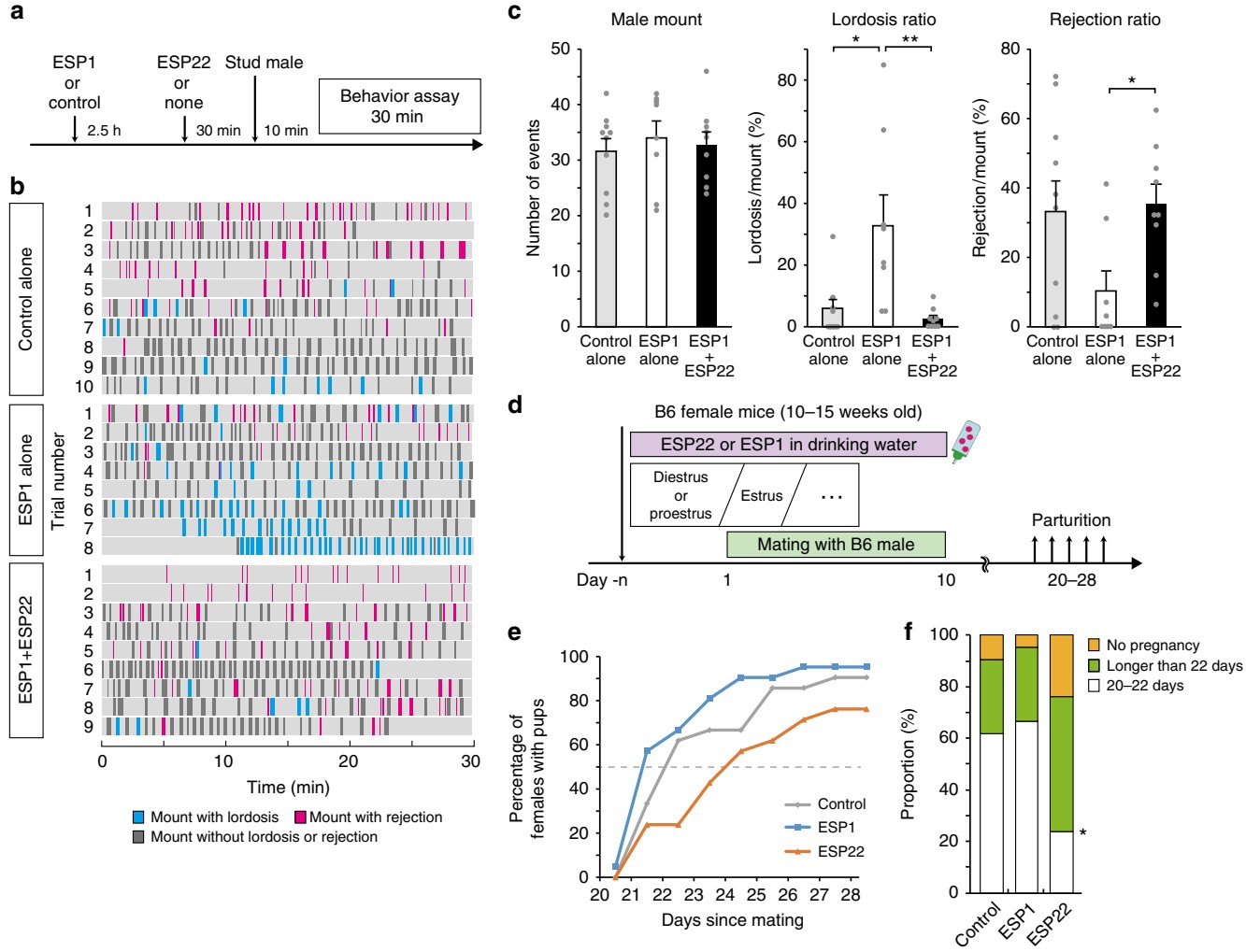

**Fig. 2** ESP22 counteracts ESP1-mediated sexual facilitation and elicits reproductive suppression. **a** Timeline for female sexual behavior assays. **b** Raster plots as detailed in Fig. 1b. Control, $n = 10$; ESP1, $n = 8$; ESP1 + ESP22, $n = 9$. **c** Quantification of the number of mounts made by males, lordosis ratio, and rejection ratio, with each gray dot representing data for an individual animal. Error bars, S.E.M. **$p < 0.01$ and *$p < 0.05$ by Steel-Dwass test. **d** Schematic setup and timeline for the reproduction assay. ESP22 or ESP1 was continuously supplied in the drinking water. Female mice at the beginning of estrus were mated for 10 days, with parturition events monitored daily. **e** A cumulative plot showing the percentage of females with pups (y-axis) with respect to the days since mating (x-axis). **f** The proportion of females in each parturition day category since mating. *$p < 0.05$ by relative risk analysis compared with control (female mice in each group, $n = 21$)

after mating was almost identical between the control-exposed and ESP1-exposed groups. In contrast, constant ESP22 exposure led to late births and a decrease in the ratio of females with pups at 20–22 days after mating (Fig. 2e, f). These data demonstrate that ESP22 acts as a reproductive suppression signal, presumably via decreased mounting of male mice[21] and receptivity of female mice (Fig. 1), that leads to a reduction in the reproduction rate of mating pairs in the vicinity.

**V2Rp4 is a functional receptor for ESP22.** Identification of ESP22 as a sexual suppression signal of female mice prompted us to identify receptor(s) for ESP22. In mice, there are 121 *V2R* genes that encode receptors for mainly non-volatile molecules[18,25]. The VNO of mice stimulated with ESP22 were analyzed by dual color in situ hybridization (ISH) with cRNA probes that recognize various *V2R* clades and an immediate early gene, *early growth response protein 1* (*Egr1*), as previous studies have established that this gene is robustly expressed by vomeronasal sensory neurons (VSNs) upon sensory stimulations[26]. A

majority (63%) of the ESP22-induced *Egr1*-positive VSNs were labeled with a cRNA probe that detected all the members of the *V2Rp* clade (Fig. 3a, b). *V2Rn* was observed in a small population (4%), with no overlap found with the other *V2R* clades (Supplementary Fig. 2a). The *V2Rp* clade contains five intact *V2R* genes[10], among which the *V2Rp4*, *V2Rp5*, and *V2Rp6* signals were observed in *Egr1*-positive neurons (Fig. 3c, d). Since these three receptor probes showed cross-hybridization due to extremely high sequence homology[10], we attempted to compare ESP22-induced neural activity of the VNO and rejection behavior in knockout mice for individual receptors. Since V2Rp5 is the receptor for ESP1 and knockout mice were available[10], we generated *V2Rp4*-deficient or *V2Rp6*-deficient mice using the CRISPR/Cas9 genome-editing system (Supplementary Fig. 2b–e)[27]. While we generated these mutant mice, it was shown that phosphorylated ribosomal protein pS6 turned out to act as a sensitive neural activity indicator in the VSNs[28]. We therefore analyzed the VNO sections from *V2Rp5*[−/−] and *V2Rp6*[−/−] mice exposed to ESP22 and found trends of increased number of cells positive for pS6. In contrast, there was no increase in the number

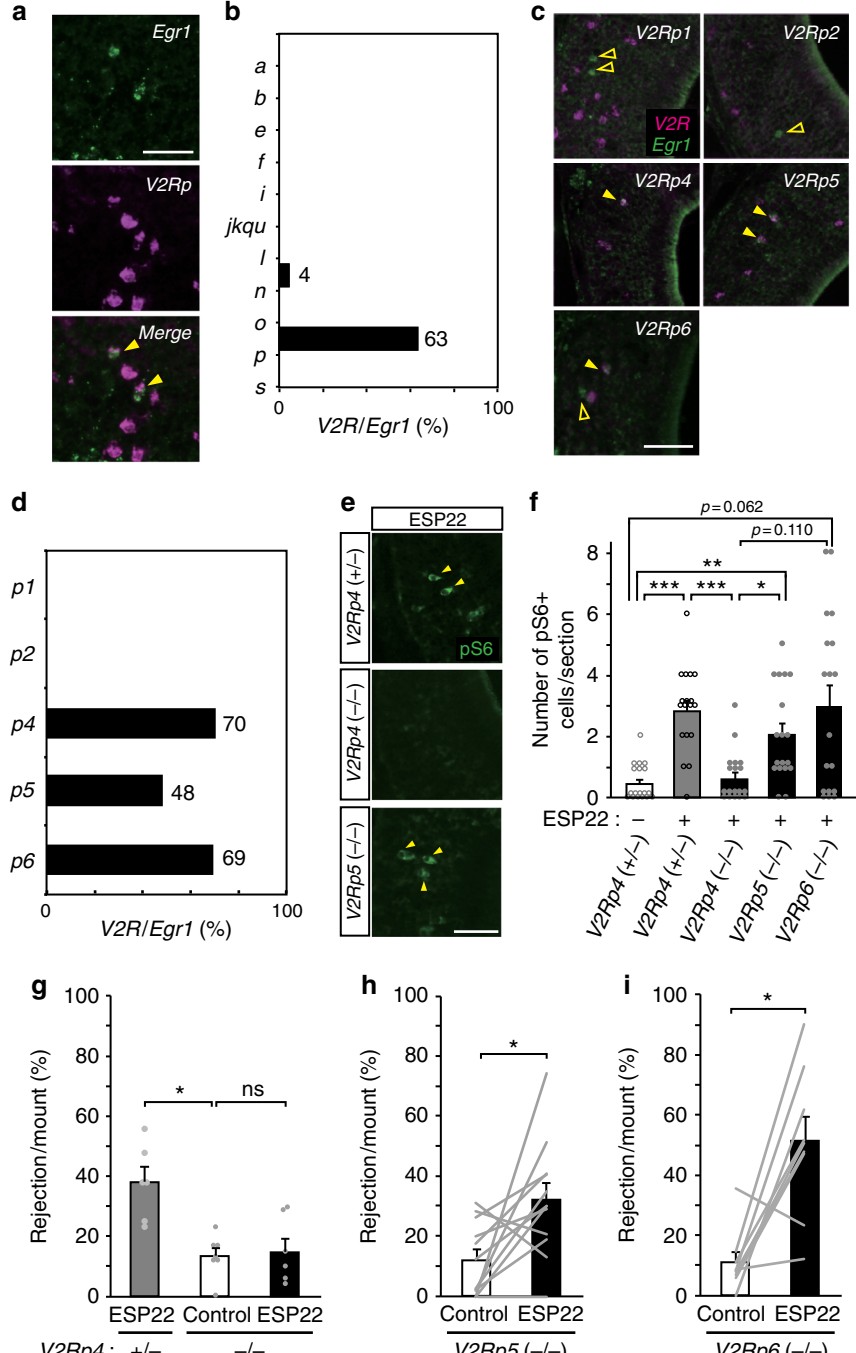

**Fig. 3** V2Rp4 is a functional receptor for ESP22. **a** Dual-color ISH staining of a VNO section from an ESP22-stimulated female mouse labeled with the *Egr1* cRNA probe (green) and *V2Rp* clade-specific cRNA probe (magenta). Closed arrowheads show double-labeled cells. Scale bar, 50 μm. **b** Percentage of *V2R*-positive VSNs among ESP22-induced *Egr1*-expressing neurons. **c** Dual-color ISH staining with *Egr1* (green) and each *V2Rp* cRNA probe (magenta). Open arrowheads show *Egr1*-positive cells; closed arrowheads show double-labeled cells. Scale bar, 50 μm. **d** Percentage of *V2Rp*-positive VSNs among the ESP22-induced *Egr1*-expressing cells. **e** Representative immunohistochemical images of pS6-expressing VSNs in ESP22-stimulated *V2Rp4$^{+/-}$*, *V2Rp4$^{-/-}$*, or *V2Rp5$^{-/-}$* mice. Arrowheads represent example pS6-positive VSNs. Scale bar, 50 μm. **f** The number of pS6-positive cells per VNO section. For each genotype, 18 sections from each of 3 animals were quantified. \*\*\*$p < 0.001$, \*\*$p < 0.01$, and \*$p < 0.05$ by Steel-Dwass test. **g–i** Quantification of the rejection ratio using *V2R* mutant female mice pre-exposed to either control buffer or ESP22. The genotype of the female mice is shown at the bottom of each graph. Gray lines and dots indicate data from an individual animal. Error bars, S.E.M. ns, not significant. \*$p < 0.05$ by Kruskal-Wallis test with Bonferroni correction in **g** or Wilcoxon signed rank test in **h**, **i**

of pS6-positive neurons found in the VNO of *V2Rp4$^{-/-}$* mice (Fig. 3e, f). We also observed a significant induction of pS6 in the VNO of *V2Rp4$^{-/-}$* mice exposed to ESP1, whereas no increase was observed in the VNO of *V2Rp5$^{-/-}$* mice (Supplementary Fig. 2f, g). These data suggest that V2Rp4 is necessary for ESP22-,

but not ESP1-, induced neural activation of VNO. A female sexual behavior assay with these vomeronasal receptor-deficient mice was also performed. Whereas *V2Rp5$^{-/-}$* or *V2Rp6$^{-/-}$* female mice in a hormone-primed pseudo-estrus state exhibited ESP22-dependent increases in rejection behavior, no such

enhancement was observed in $V2Rp4^{-/-}$ mice (Fig. 3g–i). Taken together, these findings indicate that V2Rp4 is the functional receptor for ESP22, and the ESP1 receptor V2Rp5 is dispensable for ESP22-induced rejection behavior.

**ESP22-induced *c-Fos* expression in the brain.** We next aimed to dissect the neural basis for ESP22-mediated sexual suppression of female mice. Along the vomeronasal neural circuit, the MeA, bed nucleus of the stria terminalis (BNST), and posteromedial cortical amygdaloid nucleus (PMCo) are the main structures that receive direct inputs from the AOB and, therefore, considered as third-order regions[20,29]. In addition, the medial preoptic area (MPA) and the VMH are often activated by vomeronasal ligands[10,30] and known to be involved in the regulation of female sexual behaviors. We, thus, analyzed *c-Fos* expression, a neural activity marker, in these five brain regions from virgin female mice exposed to ESP22. Among the third-order regions, ESP22 induced *c-Fos* in the MeA (most intensively in the posteroventral part of the MeA, or MeApv) and broad subdivisions of the BNST, but not in the PMCo (Fig. 4a–d and Supplementary Fig. 3a, b). *c-Fos* induction was mainly observed in steroidogenic factor 1 (SF1)-positive neurons[31] located in the VMHd, but not in the MPA (Fig. 4e, f and Supplementary Fig. 3a, b).

MeApv is traditionally thought to transmit information about defensive responses[32,33]. A recent study revealed that MeApv neurons can be divided into at least two subtypes based on their axonal projections to the BNST/MPA and VMHd, respectively[13]. To reveal whether information about ESP22 is conveyed via specific subtypes in the MeApv of female mice, we labeled neurons projecting to BNST or VMHd by injecting a retrograde tracer, Red Retrobeads, into one of these two regions and examined whether they were activated by ESP22 stimulation (Supplementary Fig. 3c). We found a significant increase in the number of Retrobeads and *c-Fos* double-labeled cells when Retrobeads was injected into the BNST (Supplementary Fig. 3d, e). This result implies that the activation of BNST by ESP22 could be, at least partly, mediated by MeApv to BNST projection neurons, as they have been shown to be glutamatergic[13].

Previous studies showed that ESP1, which has inverse effects on female sexual behavior compared to ESP22, also activated MeA to enhance female sexual behavior[10,13]. How does the same brain region (MeA) control female sexual behavior in contrary ways? To address this question, we examined how the two pheromonal signals are represented within the MeApv by using cellular compartment analysis of temporal activity with fluorescent ISH (catFISH) (Supplementary Fig. 3f)[13,34,35]. We used $NR4a1^{36}$ mRNA probes here because its nuclear transcript was robustly detected in the MeApv but intron probes of *c-Fos* easily generated non-specific labeling in our pilot experiments. In the MeApv sections from female mice that received the ESP1 stimulus twice, nearly 80% of the cells that expressed immediate early gene *NR4a1* transcripts in the nuclei also expressed cytoplasmic *NR4a1* mRNA. In contrast, in mice that serially received ESP1 and then ESP22, less than 20% of the cells exhibited nuclear and cytoplasmic transcripts (Supplementary Fig. 3g). These results indicated that ESP1 and ESP22 activated mostly non-overlapping neural populations in the MeApv. Together with the fact that ESP1 and ESP22 are received by different receptors, V2Rp5 and V2Rp4 (Fig. 3)[10], these results suggest that ESP1 and ESP22 signals are transmitted in parallel from their specific vomeronasal receptors to MeA; these parallel pathways may underlie the opposing effects of these pheromones on female sexual behaviors.

**MeA and BNST are necessary for sexual suppression by ESP22.** Next, to address the necessity of specific brain regions and the

specific cell types identified by *c-Fos* mapping during ESP22-mediated sexual rejection, we virally targeted mCherry-tagged hM4Di, a CNO (clozapine-N-oxide)-dependent neural silencer[37], to the MeA and BNST of wild type female mice by injecting a mixture of *AAV-Cre* and *AAV-FLEx-hM4Di-mCherry*. We also injected *AAV-FLEx-hM4Di-mCherry* into the VMHd of *SF1*-Cre female mice (Fig. 5a). Although we observed considerable variabilities in rejection ratio in some of sexual behavior assays using these mice, presumably due to viral injection surgery-associated stress, control groups in which saline was intraperitoneally injected before the assay still showed a trend of ESP22-induced increase of rejection ratio. In contrast, when hM4Di was properly targeted to the MeA or BNST, intraperitoneal injection of CNO impaired ESP22-dependent sexual rejection.

In the MeA, post-hoc histological analysis revealed that hM4Di-mCherry was almost evenly targeted among the anterior, posterodorsal, and posteroventral parts of the MeA (MeAa, MeApd, and MeApv, respectively) (Fig. 5b and Supplementary Fig. 4a). We, thus, analyzed the correlation between the Δrejection ratio (ESP22-control) and hM4Di expression level (inferred by mCherry expression) and found a statistically significant negative correlation ($p = 0.03$) only in the MeApv (Fig. 5c–e and Supplementary Fig. 4b). Although there was a trend towards negative correlation in the MeAa, no such correlation was found in the MeApd. In the BNST, hM4Di-mCherry expression was primarily distributed in the medial anterior (MA), lateral (L), and medial posterior (MP) areas, and a significant negative correlation between Δrejection ratio and hM4Di expression level was found when the entire BNST region was analyzed (Fig. 5f–h and Supplementary Fig. 4b). Together, these data suggest that MeA (mainly in MeApv) and BNST neurons are necessary for ESP22-mediated enhancement of sexual rejection. Because MeApv ESP22-responding neurons sent projections to the BNST (Supplementary Fig. 3c), one possible interpretation of these data is that the MeApv-BNST pathway mediates ESP22-induced responses.

In sharp contrast to results in the MeA or BNST, ESP22-dependent rejection persisted in female mice in which hM4Di was targeted to *SF1*-positive VMHd neurons (Fig. 5a, b, i–k). This pharmacogenetic data was further confirmed by a cell type-specific ablation experiment; ESP22-dependent sexual rejection was still observed in female mice in which the majority of *SF1*-positive VMHd neurons were ablated with bilateral viral injection of diphtheria toxin A (DTA) via *AAV-FLEx-DTA* (Supplementary Fig. 5a–d), suggesting their dispensability for ESP22-mediated sexual suppression.

**The inhibitory pathway sufficient for sexual rejection.** We next aimed to identify a pathway between BNST neurons and downstream targets that sufficiently mediates the sexual suppression signal in female mice. Dual-color ISH with *c-Fos* and *glutamate decarboxylase 1 + 2 (GAD1 + 2)* cRNA probes in brain sections from virgin female mice exposed to ESP22 revealed that the majority of ESP22-activated cells in the BNST were GABAergic (dual-labeled cells per *c-Fos*+ cells: $96.9 \pm 0.6\%$ in whole BNST, $98.3 \pm 1.7\%$ in MA area, $95.1 \pm 1.2\%$ in MP area, mean ± S.E.M, $n = 3$ animals) (Fig. 6a). To identify the downstream targets of these neurons, we visualized the axons of the BNST GABAergic neurons by injecting a mixture of *CAV2-FLEx-Flp* and *AAV-FLEx-mGFP-2a-SypRuby*[38,39] into the unilateral BNST of *GAD2-Cre* female mice (Fig. 6b). This enabled mGFP labeling in the axons as well as sypRuby labeling in the synaptic boutons. Over 70% of mGFP+ cells (termed source cells, green) were localized in the MA and MP regions of the BNST. A relatively small fraction was observed in the surrounding structures, such as

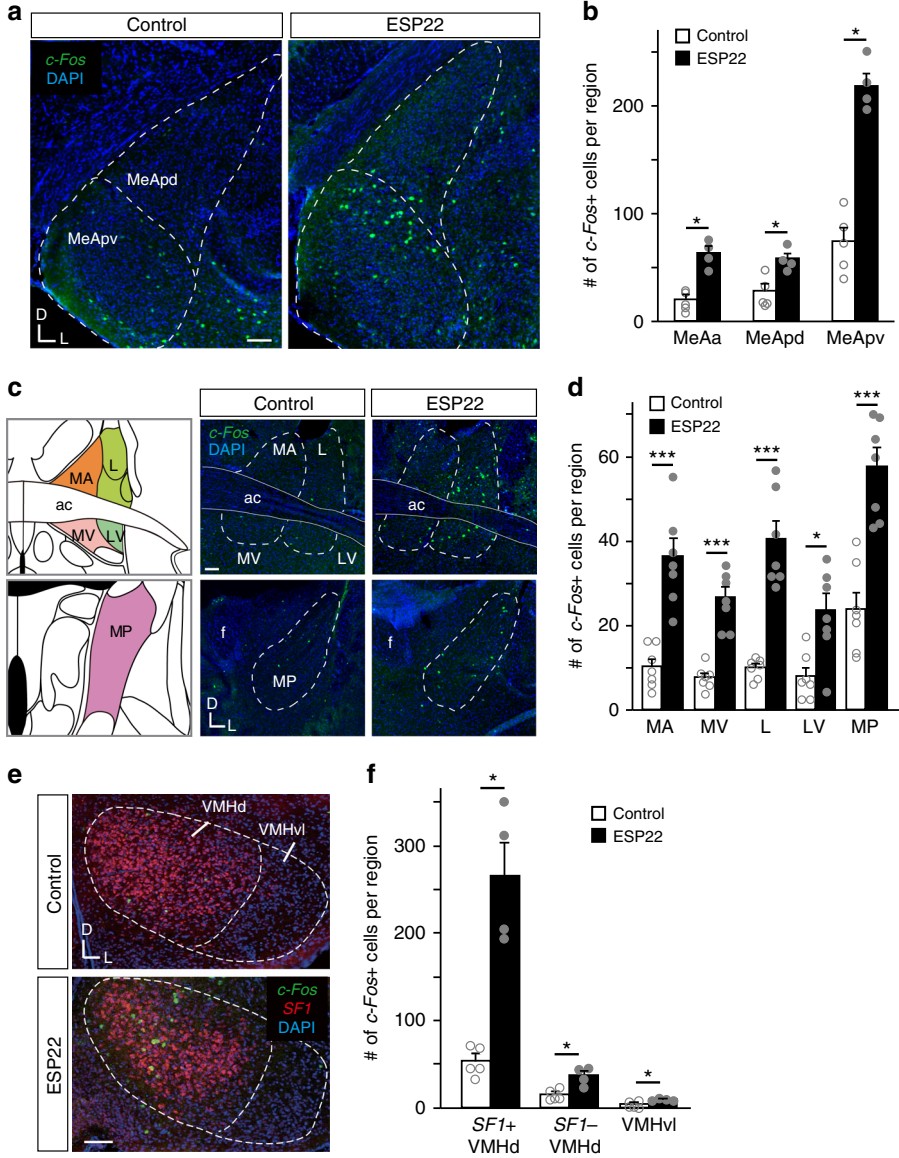

**Fig. 4** ESP22-induced *c-Fos* expression in the amygdala-hypothalamus axis of female mice. **a** Representative ISH sections of the MeA from female mice stimulated with control buffer or ESP22. *c-Fos* cRNA probe (green) was used in conjunction with nuclear DAPI staining (blue). Abbreviations: MeApd, MeA posterodorsal part; MeApv, MeA posteroventral part; D, dorsal; and V, ventral. Scale bar, 100 μm. **b** Quantification of *c-Fos*-positive neurons in the MeA. The number of sections counted to determine the number of *c-Fos*-positive neurons in each brain area of each animal were: MeAa, 4; MeApd, 12; and MeApv, 12. Error bars, S.E.M. $n = 4$-5. MeAa, MeA anterior part. **c** Representative ISH sections of the BNST, as detailed in **a**. MA, medial division anterior part; L, lateral division; MV, medial division ventral part; LV, lateral division ventral part; MP, medial division posterior part; f, fornix; and ac, anterior commissure. Scale bar, 100 μm. Image adapted from the Allen Mouse Brain Atlas (©2004 Allen Institute for Brain Science. Allen Mouse Brain Atlas. Available from: mouse.brain-map.org). **d** Quantification of *c-Fos*-positive neurons in the BNST. Ten sections were counted for each BNST area per animal. $n = 7$. **e** Representative ISH sections of the VMH from female mice stimulated with control buffer or ESP22. *SF1* cRNA probe (red) and *c-Fos* cRNA probe (green) were used in conjunction with nuclear DAPI staining (blue). Abbreviations: VMHd, VMH dorsal part; VMHvl, VMH ventrolateral part; D, dorsal; and L, lateral. **f** Quantification of *c-Fos*-expressing neurons in the VMH. $n = 4$-7; 16 sections from each animal were quantified. Error bars, S.E.M. $***p < 0.001$ and $*p < 0.05$ by Wilcoxon rank sum test

lateral septum and globus pallidus (Fig. 6c, d). Mapping of GFP+ axons showed that the MPA and VMHvl were the major output targets (Fig. 6e, f).

Since VMHvl, especially neurons expressing *estrogen receptor 1 (Esr1)* in this region, has classically been characterized to be critical for female sexual behavior[11–13], we hypothesized that these neurons are important downstream targets. Thus, we next conducted rabies virus-mediated retrograde trans-synaptic tracing[40] from *Esr1* + VMHvl neurons. First, a mixture of *AAV-FLEx-TCb* and *AAV-FLEx-RG* was injected into the VMHvl of

*Esr1-Cre* female mice. Two weeks after the AAV injection, rabies virus *ΔG-GFP* + EnvA was injected into the VMHvl to initiate trans-synaptic tracing. The results revealed that broad subdivisions of BNST were trans-synaptically labeled (shown in green in Fig. 6g) from starter cells in the VMHvl (shown in yellow). The distribution of GFP+ cells in each brain area demonstrated that BNST is a prominent pre-synaptic structure of VMHvl *Esr1* + neurons (Fig. 6h). Furthermore, we confirmed that BNST neurons that mono-synaptically project to *Esr1* + VMHvl neurons were mostly GABAergic ($97.0 \pm 0.2\%$ in whole BNST,

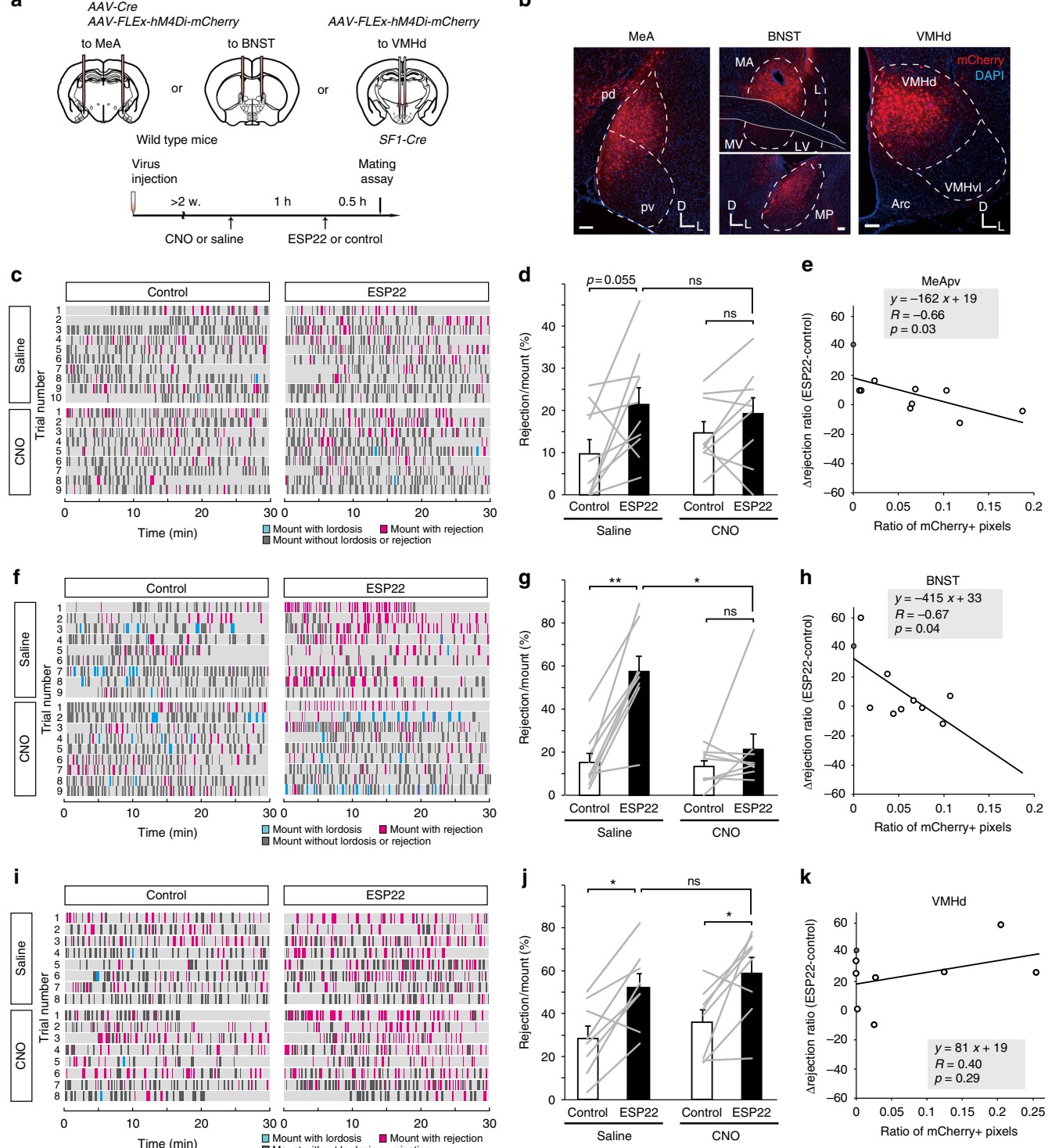

**Fig. 5** MeA and BNST neurons are necessary for female sexual suppression by ESP22. **a** Schematic illustration of the setup and timeline for loss-of-function experiments that targeted the MeA, BNST, and VMHd. Image adapted from the Allen Mouse Brain Atlas (©2004 Allen Institute for Brain Science. Allen Mouse Brain Atlas. Available from: mouse.brain-map.org). **b** Representative images of mCherry expression in the MeA, BNST, and VMHd of female mice. For abbreviations, see Fig. 4 legend. Scale bar, 100 μm. **c, f, i** Raster plots of loss-of-function experiments that targeted the MeA (**c**), BNST (**f**), and VMH (**i**) as detailed in Fig. 1b. Each animal in either the saline or CNO groups underwent two behavioral assays: one with control buffer exposure and one with ESP22 exposure. Animals that received hM4Di injection as follows: MeA (**c**), n = 10 for saline, n = 9 for CNO. BNST (**f**), n = 9 for each group. VMH (**i**), n = 8 for each group. **d, g, j** Quantification of the rejection ratio of female mice pre-exposed to control buffer or ESP22 in MeA (**d**), BNST (**g**), and VMH (**j**) groups. Error bars, S.E.M. ns, not significant. **p < 0.01 and *p < 0.05 by Wilcoxon signed rank test with Bonferroni correction. **e, h, k** Correlation between the ratio of mCherry-positive pixels (x-axis) in the MeApv (**e**), BNST (**h**), and VMHd (**k**) and the Δrejection ratio (y-axis). Open circles represent data from the hM4Di-injected CNO group, and grey circles on the y-axis represent averaged data from wild type mice (no-manipulation) to serve as a reference

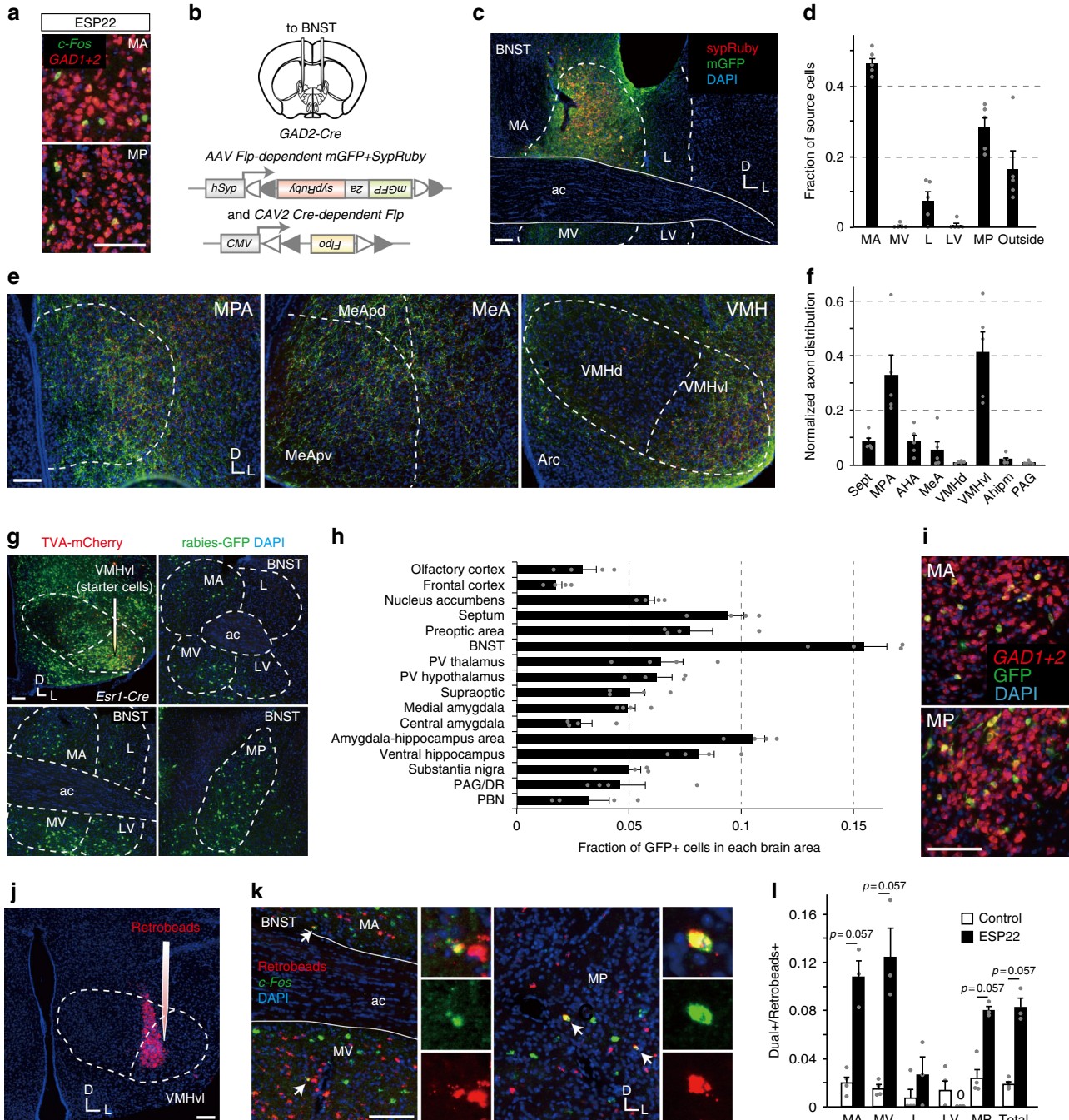

**Fig. 6** BNST GABAergic projection neurons to VMHvl are activated by ESP22. **a** Representative ISH sections of the BNST from female mice stimulated with ESP22 and stained with *c-Fos* (green) and *GAD1+2* cRNA probes (red). **b** Schematic illustration of the setup for axon mapping of BNST GABAergic neurons. Image adapted from the Allen Mouse Brain Atlas (©2004 Allen Institute for Brain Science. Allen Mouse Brain Atlas. Available from: mouse.brain-map.org). **c** Representative BNST images showing the distribution of source cells (green). **d** Fraction of the source cells among the five subdivisions of the BNST and surrounding structures. **e** Representative coronal sections of the MPA, MeA, and VMH showing axons of GABAergic neurons in the BNST (green) and their putative pre-synaptic structures (red). **f** Proportion of axonal arborization of the BNST GABAergic neurons among the eight brain regions. *n* = 5. Abbreviations: Sept, lateral septum; AHA, anterior hypothalamus; AHipm, amygdalohippocampal area posteromedial part; and PAG, periaqueductal gray. **g** Representative sections of the VMH and BNST from female mice after rabies virus-mediated trans-synaptic tracing. **h** Fraction of GFP+ cells in each brain area normalized by the total number of GFP+ cells in all 16 brain regions. *n* = 4. Abbreviations: PV, paraventricular; DR, dorsal raphe nucleus; and PBN, parabrachial nucleus. For other abbreviations, see Fig. 4. **i** BNST coronal sections labeled with GFP (green) and *GAD1+2* (red) ISH probes. **j** A representative coronal VMH section showing the injection site of red Retrobeads. **k** Representative examples of BNST ISH sections, including the MA, MV, and MP, with ESP22-induced *c-Fos*-positive (green) and red Retrobeads-positive (red) neurons labeled from the VMHvl. Arrows indicate double-positive cells, which are shown in high magnification in the merged and separated channels on the right. **l** Ratio of double-positive neurons to red Retrobeads-positive neurons in the five BNST subdivisions and total BNST (*n* = 4 for control buffer, and *n* = 3 for ESP22). Data from individual animals are represented with gray dots. Error bars, S.E.M. Statistical analysis by Wilcoxon rank sum test. Scale bar, 100 μm. D, dorsal; L, lateral

$99.4 \pm 0.6\%$ in MA region, and $96.3 \pm 1.3\%$ in MP region) (Fig. 6i).

To further confirm that VMHvl-projecting neurons in the BNST are activated by ESP22, we conducted ISH with *c-Fos* cRNA probes in the brain sections from female mice injected with red Retrobeads into the VMHvl and then stimulated with ESP22 (Fig. 6j). Ratio of Retrobeads and *c-Fos* double + neurons over Retrobeads + neurons in the total BNST and three BNST subdivisions tended to increase (Fig. 6k, l). Collectively, these data show that ESP22 activates inhibitory neurons projecting from the BNST to the VMHvl.

To directly test the sufficiency of this pathway in suppressing female sexual behavior, we performed gain-of-function experiments with optogenetic activation[41]. The optical fibers were implanted above the BNST or VMHvl after viral injection of *AAV* expressing channelrhodopsin-2 (ChR2)[41] in a Cre-dependent manner (*AAV-FLEx-ChR2*) or injection of *AAV-FLEx-GFP* as a control into the BNST of *GAD2-Cre* female mice (Fig. 7a, b and Supplementary Fig. 6). Light stimulation of BNST resulted in a significant increase in *c-Fos* expression in *ChR2*-expressing animals (*GAD2::ChR2-BNST*) compared to GFP-expressing control animals (*GAD2::GFP-BNST*) (Fig. 7c). During a mating assay, we found no enhancement of sexual rejection with photo-stimulation in trials with control virus injection (*GAD2::GFP-BNST*). By contrast, optogenetic activation of GABAergic neurons in the BNST (*GAD2::ChR2-BNST*) or their axon terminals in the VMHvl (*GAD2::ChR2-VMHvl*) for 5 min one hour before the mating assay elicited prolonged sexual rejection in female mice (Fig. 7d, e). It was also shown that light stimulation of *GAD2* + BNST axon fibers in the VMHvl does not significantly activate BNST cell bodies by back-propagating action potentials (Supplementary Fig. 7a–d). Based on all of these findings, we propose a neural circuit model in which ESP22 recruits the inhibitory BNST to VMHvl pathway, thereby resulting in sexual suppression in female mice (Fig. 7f, blue lines).

## Discussion

This study finds that (1) a chemical signal from juvenile mice, ESP22, can promote sexual rejection behavior in female mice, (2) this juvenile-to-adult communication pathway requires a dedicated ESP22 sensory receptor termed V2Rp4, and (3) ESP22-mediated sexual suppression involved a critical limbic system connection from the BSNT to the VMHvl. These findings provide key insights on how information is selectively routed through the olfactory system, and more generally provide a neuronal basis for why different sensory inputs evoke variable behaviors.

Classical studies of chemosensory signals that negatively affect female reproduction have emphasized nearby adult conspecifics to be sources of pheromones[3]. For example, exposure to group-housed adult female odors can delay the onset of puberty[42] and also suppress the estrus cycle of adult female mice[43]. Even after mating, exposure to unfamiliar adult male chemosensory signals, including ESP1, can induce a high rate of pregnancy failure[23], which is a phenomenon known as the Bruce effect[44]. In addition to these adult-derived signals, the current study demonstrates that younger and immature conspecifics are another source of pheromones that negatively influence female reproduction. Another important feature of ESP22 is that it can directly influence the choice of sexual behavior of female mice without affecting their estrus cycle. This relatively acute effect was still significant enough to reduce the reproduction rate if mice were chronically exposed to ESP22 (Fig. 2d–f). We speculate that ESP22-mediated sexual suppression could be beneficial for females in natural environments, as the presence of growing juveniles is an unequivocal signal that predicts future local resource competition and

potential unsuccessful reproduction due to overpopulation[2,4] or infanticide[45]. For juvenile mice, it might ensure fewer competitors and a resource-rich environment for improved survival. Testing these ideas must await future studies, including ones using semi-natural outdoor enclosures and specific mutant mice that are deficient in producing or sensing the pheromone.

Other major findings of this study include the functional identification of a specific receptor for ESP22 and the neural circuitry responsible for ESP22-induced behaviors by pharmaco-/opto-genetic manipulations. Interestingly, the identified neural basis for ESP22-induced behaviors (Fig. 7f, blue lines) exhibits strong parallels with that for ESP1[13], which positively influences sexual behavior of female mice (Fig. 7f, red lines). ESP22 is detected by a single receptor type, V2Rp4 (Fig. 3), while ESP1 is received by V2Rp5. These receptors are encoded by adjacent genes and are highly homologous (87% identical in amino acid sequences[10]), yet they only detect their corresponding pheromone. In the amygdala, ESP22 and ESP1 activated the same region MeApv but in mostly non-overlapping neural populations (Supplementary Fig. 3f, g). They also had distinct axonal projection pathways: V2Rp4 (ESP22) information was more preferentially targeted to the BNST pathway (Supplementary Fig. 3c–e), while V2Rp5 (ESP1) information was biased toward the VMHd pathway[13]. Genetic loss-of-function experiments of MeA activities tended to impair both ESP22- (sexual suppression) and ESP1- (sexual enhancement) induced behaviors. Thus, we propose that MeA acts as a hub to route vomeronasal inputs of varying functions to their appropriate downstream neural circuits. Downstream of the MeA, BNST, but not VMHd, is required for ESP22 processing, whereas VMHd, but not BNST, is required for ESP1 processing (Fig. 5)[13]. ESP22 induced activations of GABAergic neurons located in broad subdivisions of BNST including anteromedial and posteromedial areas, and they convey inhibitory input to the VMHvl, one of the most critical brain areas for female sexual behavior[11,12,46]. This circuit organization can account for ESP22-induced negative influence on female reproduction, as supported by optogenetic stimulation experiments (Fig. 7). Taken together, our studies demonstrate that different pheromones (ESP22 or ESP1) and, thus, different vomeronasal receptors (V2Rp4 or V2Rp5), couple with distinct neural circuits. Future studies will, hopefully, reveal the cellular and molecular mechanisms underlying the formation of dedicated neural circuits downstream of specific vomeronasal receptors.

## Methods

**Animals**. Animals were housed under a regular 12 h dark/light cycle with food and water *ad libitum*. Wild type ICR male mice and C57BL/6N mice were purchased from Japan SLC (Tokyo, Japan). Wild type C57BL/6J mice were purchased from Japan SLC for histological experiments and from Japan CLEA (Shizuoka, Japan) for behavioral experiments. C3H/HeJ mice were purchased from Japan SLC. V2Rp5-deficient mice were generated previously[10]. V2Rp4- and V2Rp6-deficient mice were generated as described in the Generation of mutant mice by CRISPR-mediated genome editing section. SF1-Cre (also known as Nr5a1-Cre, Jax#012462), GAD2-Cre (Jax#010802), and Esr1-Cre (Jax#017911) mice were purchased from The Jackson Laboratory (Bar Harbor, Me, USA). Experiments were carried out in accordance with the animal protocols approved by the Animal Care and Use Committees at the University of Tokyo and RIKEN.

**Generation of mutant mice by CRISPR-mediated genome editing**. To generate the null mutant of V2Rp4 (a.k.a. Vmn2r115) or V2Rp6 (a.k.a. Vmn2r117), we designed two guide RNAs (gRNAs) for each gene that were able to introduce double-strand DNA breaks flanking exon 6 of the V2R gene in which the essential transmembrane domain is encoded. Cas9 mRNA was prepared as follows[47]. pMLM3613 (Addgene, #42251) was digested with PmeI and purified with ethanol precipitation. In vitro transcription was performed using mMESSAGE mMACHI NE T7 ULTRA Transcription Kit (ThermoFisher Scientific, #AM1345) in accordance with the manufacturer's instructions. The amount and purity of synthesized mRNA were tested using electrophoresis with a 1% agarose gel. To design the gRNAs used to target V2Rp4 or V2Rp6, we first searched for 20 bp target sequences

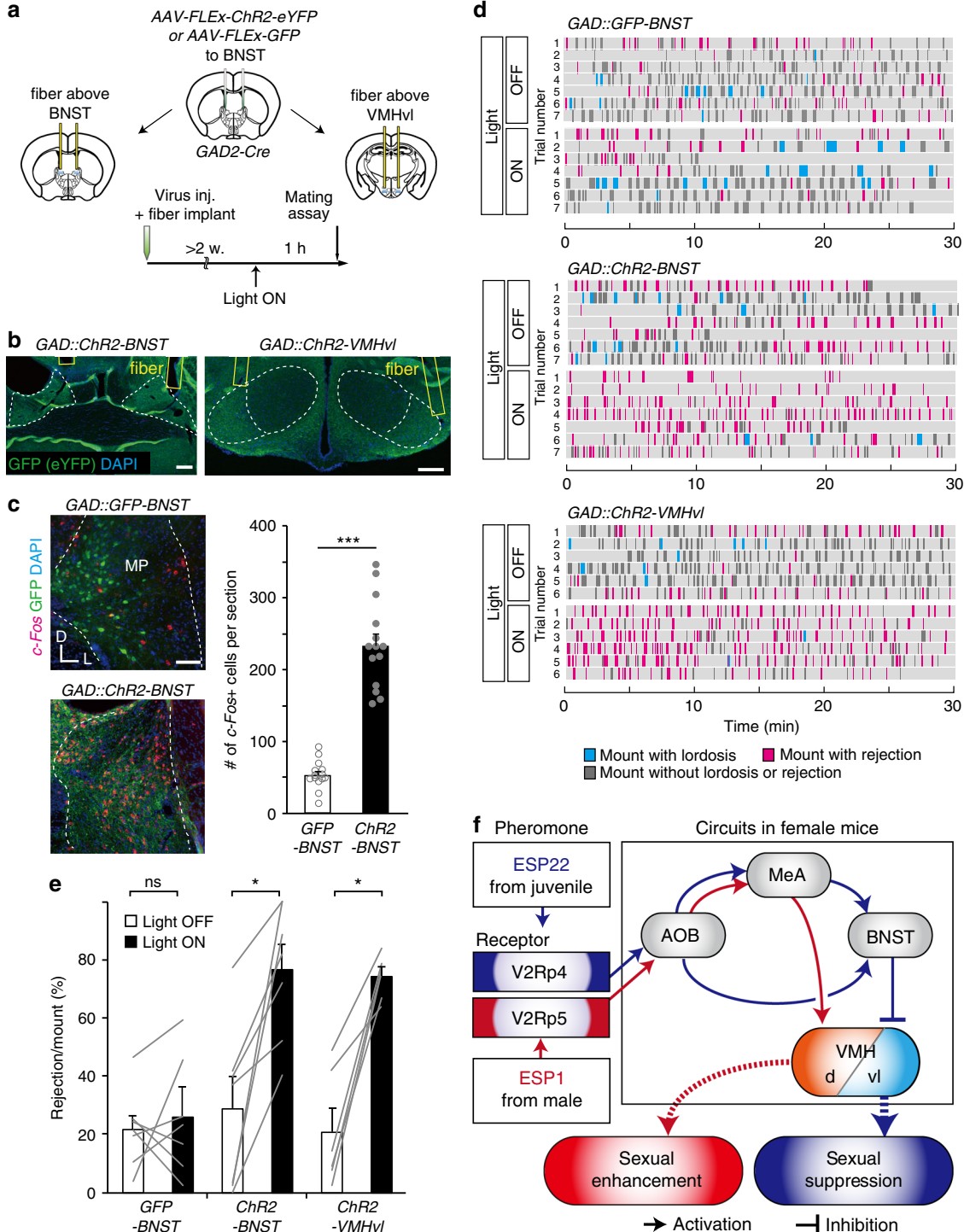

**Fig. 7** Optogenetic activation of axons from *GAD2*-positive BNST neurons in the VMHvl elicited sexual rejection. **a** Schematic illustration of the setup and timeline for gain-of-function experiments that targeted *GAD2*-positive BNST neurons by optogenetic activation of their cell bodies in the BNST or their axons in the VMHvl. Image adapted from the Allen Mouse Brain Atlas (©2004 Allen Institute for Brain Science. Allen Mouse Brain Atlas. Available from: mouse.brain-map.org). **b** Representative coronal section showing location of fiber tip in the BNST (left) or VMHvl (right). Scale bar, 200 μm. **c** Representative ISH sections showing *c-Fos*-positive cells (red) induced by photoactivation and GFP-positive (ChR2-YFP expression) cells (green) in the BNST of *GAD2::ChR2-BNST* or *GAD2::GFP-BNST* animals. Quantification of the number of *c-Fos*-positive cells per section is shown in the graphs. ***$p < 0.001$ by Wilcoxon rank sum test. Scale bar, 100 μm. **d** Raster plots as detailed in Fig. 1b for female mice expressing ChR2 in BNST GABAergic neurons that received photoactivation in the BNST (*GAD2::ChR2-BNST*, $n = 7$) or VMHvl (*GAD2::ChR2-VMHvl*, $n = 6$), as well as for control female mice expressing GFP in BNST GABAergic neurons that received photoactivation in the BNST (*GAD2::GFP-BNST*, $n = 7$). **e** Quantification of the rejection ratio for optogenetic experiments. Error bars, S.E.M. ns, not significant. *$p < 0.05$ by Wilcoxon signed rank test. **f** Model of neural pathways responsible for ESP22-mediated sexual suppression of female mice (blue lines) and ESP1-mediated enhancement of sexual receptivity (red lines)[13]

upstream of the protospacer adjacent motif (PAM) using CRISPRdirect (http://crispr.dbcls.jp/). We then selected a target sequence with >50% CG content that was completely unique in the mouse genome (confirmed by GGGenome at http://gggenome.dbcls.jp/ja/mm10/2/). The selected sequence was then introduced into the BsaI-digested pDR274 construct (Addgene, #42250) using the following oligo-DNAs: V2Rp4 upstream

5′-TAGGGATGGCTAATGAGCGGTGTT; 5′-AAACAACACCGCTCATTAGCCATC.

V2Rp4 downstream

5′-TAGGTGTCCCAGGTATGGGCATAC; 5′-AAACGTATGCCCATACCTGGGACA.

V2Rp6 upstream

5′-TAGGACCTGTCAGGGCTGGCTAGT; 5′-AAACACTAGCCAGCCCTGACAGGT.

V2Rp6 downstream

5′-TAGGTCCCCAAGGTCTCCAATGGT; 5′-AAACACCATTGGAGACCTTGGGGA.

After validating the sequence, pDR274 with the target DNA sequence was digested with DraI, and in vitro transcription of gRNA was performed using the MEGAshortscript T7 Transcription Kit (ThemoFisher Scientific, #AM1354) in accordance with the manufacturer's instructions. The synthesized gRNA was purified using the MEGAclear Transcription Clean-Up Kit (ThemoFisher Scientific, #AM1908). The amount and purity of the synthesized gRNA were tested using electrophoresis with a 1% agarose gel. A mixture of 20 ng μl$^{-1}$ of two gRNAs and 50 ng μl$^{-1}$ of Cas9 mRNA was injected into C57BL/6J fertilized eggs in order to generate the knockout mice. The genotypes of the mutant mice were determined by PCR using the following oligo-DNAs. For V2Rp4: 5′-GCATTCTAACCAACCTGTTGAG;

5′- GCCAAGCGTTTTGATGGCCAAAATTGCAC.

Mutant allele = 548 bp. Wild type allele = no amplification.

For V2Rp6: 5′-TCCATGATAAATGGCCAAAACAACA;

5′-TAGACTGGCTAGGTGGC; 5′-GAAGGAAGGCCCAAAGTCAA.

Mutant allele = 566 bp. Wild type allele = 470 bp.

The efficiency of the CRISPR-mediated gene knockout was as follows: V2Rp4, 1 out of 36 pups (2.8%); V2Rp6, 3 out of 26 pups (11.5%). As the efficiency of the V2Rp4 knockout was low, we additionally designed two upstream gRNAs for V2Rp4 (one upstream of exon 5 and one upstream of exon 6) using the following oligo-DNAs.

V2Rp4 upstream 2

5′-TAGGTGTTGGAAACCTGATTGCCA;

5′-AAACTGGCAATCAGGTTTCCAACA.

V2Rp4 upstream 3

5′-TAGGTGTCCCAGGTATGGGCATAC;

5′-AAACGTATGCCCATACCTGGGACA.

When the V2Rp4 upstream 2 + V2Rp4 downstream gRNAs were used, genotyping PCR was conducted using the following primers:

5′-TGGTTTTACATAAGGCCATGCT;

5′-CCAAGCGTTTTGATGGCCAAAATTGCAC.

Mutant allele = 480 bp. Wild type allele = no amplification. Using this procedure, we were able to obtain 8 out of 63 pups (12.7%) with a deletion of exon 6 of V2Rp4.

When V2Rp4 upstream 3 + V2Rp4 downstream gRNAs were used, genotyping PCR was conducted using the following primers: 5′-CTTGTACATGATAACTGCCAGTTGT;

5′-GCCAAGCGTTTTGATGGCCAAAATTGCAC.

Mutant allele = 470 bp. Wild type allele = no amplification. Using this procedure, we were able to obtain 6 out of 52 pups (11.5%) with a deletion of exon 5–6 of V2Rp4. We established mutant mouse lines from these F0 founders and used the F2–F5 generations for our experiments.

**Production of recombinant ESP22.** Total RNA was prepared from the external lacrimal gland of juvenile C57BL/6 mice using TRIzol reagent (Invitrogen). After DNase I (Promega) treatment, total cDNA was synthesized using Superscript III Reverse Transcriptase (Invitrogen). ESP22 cDNA without signal sequences (corresponding to Arg$^{23}$-Thr$^{111}$) was obtained by RT-PCR using the following primers:

5′-GAAGGAGAAGGAGAACATATGATGGTTCTGAAACAGACTCAA;

5′-GAAGGACTCGAGTCCATCCTCCTCATCGACTCTAA.

The amplified DNA was subcloned into the expression vector pET-28a (Novagen). The expression construct was transformed into E. coli BL21 (DE3). Expression of the peptide was induced with isopropyl thiogalactoside for 4 h. Bacterial pellets were resuspended in 1× binding buffer (0.5 M NaCl, 20 mM Tris-HCl, 40 mM imidazole, pH 7.9) and sonicated. Bacterial pellets from 50 ml LB buffer contained about 1 mg ESP22. After centrifugation, the bacterial pellets were resuspended in 1× binding buffer with 6M urine and purified using the His-Bind purification kit (Novagen, #69864) in accordance with the manufacturer's protocol. After purification, the supernatant was applied to a reverse-phase C4 column (Sensyu) in order to exclude imidazole from the elution buffer. The fraction with significant absorbance at 220 nm was collected and freeze-dried using a freeze dryer (Tokyo Rikakikai, EYELA FDU-2200). ESP22 was resuspended in 20 mM

Tris-HCl (pH 7.5) and stored at −80°C before use. SDS-PAGE electrophoresis of the peak fraction was carried out to confirm the purity and size of the peptide.

**Behavior assays.** Female sexual behavior assay (Figs. 1, 2a–c, 3g–i, 5, 7a–e, Supplementary Figs. 1e–g, and 5) was conducted as follows[10,13]. Female mice with natural sexual cycles were primed with 0.1 ml estrogen (Wako Pure Chemical Industries, 0.4 mg ml$^{-1}$ in corn oil) 24 and 48 h before the assay and with 0.05 ml progesterone (Wako Pure Chemical Industries, 10 mg ml$^{-1}$ in corn oil) 4 h before the assay to mimic their phase of estrus during the assay. In addition, handling treatment was performed at least 10 days to keep their calmness. Female mice were isolated two days before the assay. For the experiments described in Fig. 3g–i, we used V2Rp4, V2Rp5, and V2Rp6 mutant female mice. For the experiments described in Figs. 5, 7a–e and Supplementary Fig. 5, the mice underwent surgeries for AAV injection, cannula implantation, and/or fiber optic implantation over two weeks prior to the sexual behavior assay. In addition, the mice underwent intra-peritoneal injection of CNO or saline, light-mediated activation of ChR2-expressing neurons, or pheromone exposure as described in the experimental setup. A trained ICR male mouse (Japan SLC) was used as a stud male. Starting at 10 min after the entry of the stud male mouse, we videotaped the subsequent 30 min for analysis of sexual behaviors. We counted the total number of mounting episodes by the male and lordosis responses by the female[10,13]. A rejection response was defined as the female mouse assuming at least one of the following postures; standing, crouching down, keeping their limb tight, or turning their body, to avoid insertion during the mounting attempt of the male mouse. We counted a single rejection event if the female mouse took these postures more than once during a single mounting attempt of the male mouse. Intromission was defined as insertion by the male that continued longer than 6 s. Of note, experimenters were blinded as to the type of stimulation, drug condition, or AAV type during behavioral annotation.

For the female behavior assay, after exposure to the pheromone, a piece of cotton (30 mg) transfused with 30 μg of ESP1 or 50 μg of ESP22 in 20 mM Tris-HCl (pH 7.5) was placed into the home cage of the female mouse in accordance with the timing described for each of the experimental settings. The concentration of ESP22 in juvenile tear fluid was 300–500 nM[21]. Since an adult mouse secretes about 200 μl of tear fluid per day, we estimated that a juvenile's tear fluid would be about half that of an adult's. In addition, since the average number of pups per birth is about 6–8, the total volume of juvenile tear fluid is estimated to be about 600–800 μl per day. Based on these estimations, about 1.8–4 μg of ESP22 is secreted from juvenile mice per day. The specific activity of rESP22 was estimated to be 1/10 to 1/25 of that of naturally secreted ESP22 in juvenile tear fluid, based on the case of rESP1[23]. Thus, the 50 μg of rESP22 utilized in this study roughly corresponds to 2–5 μg of natural ESP22, and this volume was not too large to mimic natural conditions.

In the experiments described in Fig. 1d–f, two C57BL/6J (ESP22+) or C3H/HeJ (ESP22-) juvenile mice (15–19 days of age) were introduced into the home cage of a female mouse for 30 min until the entry of the stud male mouse. In the experiments described in Supplementary Fig. 1e–g, C57BL/6J lactating mothers with four to six C57BL/6J (ESP22+) or C3H/HeJ (ESP22−) juvenile mice were used. Juvenile mice were removed from their home cages 30 min before the entry of the stud male mouse.

Of note, robust sexual rejection was observed in experienced female mice when BNST GABAergic neurons (or their axon termini in the VMHvl) were opto-genetically activated in the experiments described in Fig. 7. Thus, the function of BNST-VMHvl circuity seems to be conserved in experienced females as well.

Social behavior assay (Supplementary Fig. 1c–d) by using the three chamber was conducted as follows[24]. Vaginal smears from female mice were assessed daily in order to determine their estrus conditions. Female mice in the estrus and diestrus phase were used in the test. The three-chamber apparatus was constructed from acrylic resin and had small square openings that allowed the animals to move from the center chamber into the left and right chambers. Each side room had a small columnar grid box (Shinfactory, Fukuoka, Japan). Animals were first habituated to the apparatus for 10 min. An unfamiliar ICR male mouse was then placed in either the left or right chamber enclosure in the grid box. The other grid box was kept empty throughout the remainder of the test. The female mouse was then allowed to freely explore all three chambers for another 10 min. We manually counted the amount of time that the female mouse investigated each grid box.

In reproduction assay (Fig. 2d–f), vaginal smears of C57BL/6N female mice (10–15 weeks old) were assessed daily in order to determine their estrus conditions. 10 μg ml$^{-1}$ ESP1 or ESP22 was continually supplied in their drinking water. Tris-HCl buffer was used as a control. During the first estrus phase after the start of the assay, each female mouse was moved to another cage for the purpose of mating with a C57BL/6N male mouse. After the mating, ESP1 or ESP22 was maintained in their drinking water for an additional 10 days. The male mouse was removed from the cage 10 days since mating, and the female mouse was observed daily for parturition until 28 days since the mating.

**Histochemistry.** To prepare the sections for ISH and immunohistochemistry, mice were anesthetized with a lethal amount of sodium pentobarbital, sacrificed, and perfused with phosphate-buffered saline (PBS) followed by 4% paraformaldehyde (PFA) in PBS. Snouts and brain tissues were post-fixed with 4% PFA in PBS

overnight. To prepare sections of the VNO (Fig. 3a–f and Supplementary Fig. 2f, g), the snouts were decalcified in 0.5 M EDTA (pH 8.0) for 48 h at 4 °C. The tissues were then cryoprotected with 30% sucrose solution in PBS at 4 °C for 24–48 h. After collecting 14 μm coronal sections of the VNO or 20–30 μm coronal sections of the brain using a Cryostat (model #CM1860, Leica), the sections were placed on MAS-coated glass slides.

Dual-color ISH of VNO (Fig. 3a, c and Supplementary Fig. 2a) and brain sections (Figs. 4e, 6a, i) was conducted as follows[13,22]. DIG-labeled probes for various V2Rs, and each member of the V2Rp clade, c-Fos, vGluT1, vGluT2, GAD1, and GAD2 were previously characterized[10,13]. ISH probes were prepared by in vitro transcription with DIG- (#11277073910) or Flu (#11685619910)-RNA labeling mix and T3 RNA polymerase (#11031163001) in accordance with the manufacturer's instructions (Roche Applied Science). Coronal sections containing the VNO or target brain region were subjected to ISH at 60 °C overnight unless otherwise noted. After the series of post-hybridization washing and blocking, DIG-positive cells were visualized with horseradish peroxidase-conjugated anti-DIG antibody (Roche Applied Science, #11207733910, 1:500 in blocking buffer) and TSA-plus Cyanine 3 (PerkinElmer, NEL744001KT, 1:70 in 1× plus amplification diluent). Flu-positive cells were visualized with anti-Flu antibody (PerkinElmer, NEF710001EA, 1:250 in blocking buffer) followed by TSA-plus biotin (PerkinElmer, NEL749A001KT, 1:70 in 1× plus amplification diluent) and streptavidin-Alexa Fluor 488 (Life Technologies, 1:250). Sections were counterstained with 4′,6-diamidino-2-phenylindole dihydrochloride (DAPI, Sigma-Aldrich, #D8417) in order to visualize the nuclei and then mounted with cover glass using Fluoromount (Diagnostic BioSystems, #K024).

Rat antiserums were raised against synthetic peptides specific to V2Rp4 or V2Rp6; V2Rp4: NH2-C(Ahx) + NARSS-COOH, V2Rp6: NH2-C + KYNSFIHLS-COOH. The antibodies were then affinity purified using affinity columns (Sulfolink) conjugated with the synthetic peptide. For immunohistochemistry of the VNO, sections were washed with Tris-buffered saline (TBS) containing 0.1% Triton X-100 (TBST) for 10 min and treated with 3% bovine serum albumin (BSA, Sigma-Aldrich) in TBST for 30 min at room temperature for blocking. Sections were then incubated with anti-pS6 antibody (Cell Signaling, #4858, 1:200 in BSA-TBST), anti-V2Rp4 (1:20 in BSA-TBST), anti-V2Rp6 (1:100 in BSA-TBST), anti-mCherry (Life Technologies, #M11217, 1:500–1:1000 in BSA-TBST), or anti-GFP (Aves Labs, #GFP-1020, 1:500 in BSA-TBST) antibody for 24 h at 4 °C. After washing three times with TBST (10 min each), the samples were incubated with Alexa Fluor 488-conjugated goat anti-rabbit IgG (Life Technologies, A11034, 1:200 in BSA-TBST) or Alexa Fluor 488-conjugated donkey anti-rat IgG (Life Technologies, A21208, 1:200 in BSA-TBST) for 1 h at room temperature. Sections were washed once with PBST for 10 min, treated with PBS containing DAPI for 20 min, rinsed with PBS, and mounted with cover glass using Fluoromount (Diagnostic BioSystems, #K024).

For screening of the ESP22 receptors (Fig. 3a–f) and c-Fos mapping (Fig. 4 and Supplementary Fig. 3a, b), adult C57BL/6N male mice only used for receptor screening or C57BL/6J female mice were singly housed for a week. A piece of cotton (30 mg) transfused with 50 μg of ESP22 in 20 mM Tris-HCl (pH 7.5) was placed into the home cage of the female mice 20–30 min before sacrifice. In the MeA and BNST, every second coronal section (30 μm) was analyzed for c-Fos mRNA expression using the single-color FISH method. In the VMH, every second coronal section (30 μm) was stained with SF1 (red) and c-Fos (green) using the two-color ISH method.

To assess c-Fos induction in Retrobeads-positive neurons in the MeA (Supplementary Fig. 3c–e) or BNST (Fig. 6j–l), 8-week-old C57BL/6J female mice were singly housed following red Retrobeads injection into the BNST, VMHd, or VMHvl. Mice were exposed to ESP22 or control buffer 30 min before sacrifice. Only animals with successful injections of Retrobeads in the target region were used for further analysis. Every third coronal section of the MeA or BNST (30 μm) was analyzed for Retrobeads and c-Fos mRNA expression using single-color ISH.

We conducted catFISH analysis (Supplementary Fig. 3f–g) as follows[13]. Eight-week-old C57BL/6J mice were singly housed for a week, and two serial stimulants were applied to their home cages for 5 min each separated by 40 min. The mouse was quickly anesthetized and perfused. 30-μm coronal sections of the MeA were analyzed by dual color ISH method with NR4a1 coding and intron probes. When the same stimulant was given twice (e.g., "ESP1-ESP1"), the nuclear transcripts positive cells (the second-stimulant responding cells) were reliably labeled with cytoplasmic mRNA (the first-stimulant responding cells), but many mRNA positive cells were not labeled with nuclear transcripts. Habituation to the same stimulation during the second application period, or less stable nature of induction of nuclear transcript may account for this observation. Whatever the mechanisms, the number of dual positive cells divided by the nuclear transcript positive cells (A-index) more robustly represents overlap in the neural representations, than the denominator having the total responding cells. Notably, previous studies also used the same index[35,48].

Sections were imaged with an Olympus BX53 microscope (4× or 10× objective) equipped with an ORCA-R2 cooled CCD camera (Hamamatsu Photonics). Images were processed in ImageJ (National Institutes of Health) and Adobe Photoshop CS2 or CS4 (Adobe Systems).

**Stereotactic surgery.** Targeting of AAV or Retrobeads into the target brain region was conducted as follows[13]. Stereotactic coordinates were defined based on the

brain atlas[49]. After mice were anesthetized with 65 mg kg$^{-1}$ ketamine (Daiichi-Sankyo) and 13 mg kg$^{-1}$ xylazine (Sigma-Aldrich) via intraperitoneal injection, they were head-fixed to stereotactic equipment (Narishige). We then injected 150–200 nl of AAV (depending upon the experimental design, a mixture of AAVs and CAV2s was used) or 200–250 nl of red Retrobeads (Lumafluor) into the target brain region at a speed of 50–100 nl min$^{-1}$ using an UMP3 pump regulated by a Micro-4 device (World Precision Instruments). The following coordinates were used (distance in mm from the Bregma for the anterior [A], posterior [P], and lateral [L] positions and from the brain surface for the ventral [V] position): BNST, A 0.4 L 0.7 V 3.6; MeA, P 0.9 L 2.1 V 4.95; VMHd, P 1.0 L 0.25 V 5.25; and VMHvl, P 1.1 L 0.55 V 5.50. After the injection, the incision was sutured, and the animal was returned to the home cage until the beginning of the behavior assay.

The following viruses were used in each experiment. The viral titer of AAV was estimated based on quantitative PCR and is presented as genome particles (gp) per ml.

DREADD-Gi loss-of-function (Fig. 5)
[MeA and BNST] 1:10 mixture of AAV serotype 8 hSyn-GFP-Cre (4.9 × 10$^{12}$ gp ml$^{-1}$) and AAV serotype 8 hSyn-DIO-hM4D(Gi)-mCherry (5.3 × 10$^{12}$ gp ml$^{-1}$) obtained from the UNC viral core.
[VMHd] AAV serotype 8 hSyn-DIO-hM4D(Gi)-mCherry (5.3 × 10$^{12}$ gp ml$^{-1}$) obtained from the UNC viral core.
Cell-type specific ablation in VMHd (Supplementary Fig. 5)
AAV serotype 8 hSyn-DIO-eGFP (8 × 10$^{12}$ gp ml$^{-1}$) obtained from the UNC viral core or AAV serotype 5 CAG-FLEx-DTA (4.5 × 10$^{12}$ gp ml$^{-1}$)[13].
Optogenetic stimulation of BNST (Fig. 7)
AAV serotype 9 EF1a-DIO-hChR2(H134R)-EYFP (1.972 × 10$^{13}$ gp ml$^{-1}$) obtained from the UPenn viral core.
Axon mapping of BNST GABAergic neurons (Fig. 6b)
1:5 mixture of CAV2-FLEx-Flp (4.5 × 10$^{12}$ gp ml$^{-1}$)[39,50] purchased from Plateforme de Vectorologie de Montpellier (PVM, France) and AAV serotype 9 hSyn-FLEx(FRT)-mGFP-2a-SypRuby (3.6 × 10$^{12}$ gp ml$^{-1}$)[39] produced by the UNC viral core.
Rabies virus-mediated retrograde trans-synaptic tracing (Fig. 6g)
1:3 mixture of AAV serotype 2 CAG-FLEx-TCb (1.2 × 10$^{13}$ gp ml$^{-1}$) and AAV serotype 2 CAG-FLEx-RG (2.4 × 10$^{12}$ gp ml$^{-1}$)[51] were produced by the UNC viral core.
RVdG-GFP used in Fig. 6g was prepared by using B7GG and BHK-EnvA cells (originally gifted by Ed Callaway)[50]. The estimated titer of RVdG-GFP + EnvA was 1 × 10$^{9}$ infectious particles ml$^{-1}$ based on serial dilutions of the virus stock followed by infection of the HEK293-TVA800 cell line.

**Neural manipulation experiments.** AAV-injected female mice underwent handling for a week and were hormone primed as described in the Female sexual behavior assay section. Soon after the first estrogen priming, the female mice were singly housed until behavioral testing.

For pharmacogenetic inactivation (Fig. 5 and Supplementary Fig. 4), at 1 h prior to pheromone stimulation (1.5 h prior to the start of mating), 0.25 ml of 0.1 μg ml$^{-1}$ clozapine-N-oxide (CNO) dissolved in saline (Sigma-Aldrich) or 0.25 ml of saline as a control was intraperitoneally injected into mice. Each animal in the CNO or saline group underwent two rounds of the sexual behavior assay and were pre-exposed 30 min prior to mating with either ESP22 or Tris buffer control. Experimenters were blind to the experimental conditions during behavioral annotation. We previously validated that hM4Di efficiently suppressed pheromone-induced neural activity in the medial amygdala[13]. After the behavioral assay, the animals were sacrificed, and the target brain regions were analyzed for mCherry fluorescence (indication of hM4Di expression) in 30 μm coronal sections.

For Cell-type specific ablation of SF1 neurons (Supplementary Fig. 5)[13], each animal in the DTA or GFP control group underwent two rounds of the sexual behavior assay pre-exposed 30 min prior to mating with either ESP22 or Tris buffer control. Experimenters were blind to the experimental conditions during behavioral annotation. After the behavioral assay, the animals were sacrificed, and 30 μm coronal sections containing the VMH were labeled with SF1 cRNA probes by ISH to analyze the number of SF1-positive neurons that remained in the VMH.

In neural activation experiments using ChR2 (Fig. 7a–e and Supplementary Fig. 6), AAV9 EF1a-DIO-hChR2(H134R)-EYFP (UPenn viral core, Cat#0872) was bilaterally injected into the BNST using the following coordinates: A 0.3, L 0.75, V 3.6.

For fiber implantation, a 0.39-NA, 200 μm core multimode fiber (Thorlabs, Cat#FT200UMT) was inserted into a 1.25 mm ferrule (Thorlabs, Cat#CFLC230-10). Fiber tips were polished using a connectorization kit (Thorlabs, Cat#CK03) and evaluated for light power at the fiber ending using a power meter (Thorlabs). Fiber optic cannulae were bilaterally implanted following a viral injection aimed 300–500 μm above the BNST or VMHvl using the following coordinates: A 0.3, L 0.75, V 3.1 for BNST, P 1.0, L 0.75, V 5.2 for VMHvl. Cannulae were permanently fixed to the skull using a layer of Super Bond C&B (Sun Medical) followed by Unifast2 (GC Corporation). For light stimulation (light ON), cannulae were connected to a 473 nm laser (Changchun New Industries) through a rotary joint (Doric, Cat#FRJ_1 × 2i_FC-2FC_0.22) and 0.22-NA patch cord (Doric, Cat#MFP_200/230/900-0.22_0.5m_FC-ZF1.25). Light (3.0 mW, 20 Hz, 20 ms pulse, 30 s) was delivered five times, with a 30 s off interval. For the controls (light OFF), cannulae were connected to the laser for 5 min with no light delivery. Each

animal was checked post-hoc to determine if the viral injection and fiber placement were correctly conducted.

Verification of ChR2-mediated activation was conducted by checking *c-Fos* expression after the light stimulation. For *GAD2::ChR2-BNST* or *GAD2::GFP-BNST* groups, animals were isolated for 1 week for habituation. Each animal received light stimulation (3.0 mW, 20 Hz, 20 ms pulse, 25 s) five times with a 25 s light off interval at 30 min before sacrifice. Light was controlled through an Arduino (ARDUINO ZERO #ABX00003) and a simple custom-made code. This code is available upon request.

**Quantification and statistical analysis.** Data were presented as mean ± S.E.M. unless otherwise mentioned. The statistical details of each experiment, including the statistical tests used, exact value of n, and what n represents, are detailed in each figure legend. Wilcoxon rank sum test was used in Figs. 4b, d, f, 6l, 7c, Supplementary Figs. 1b, 3b, e, and 7d, with multiple comparisons with Holm correction additionally used in Fig. 1c, f, and Supplementary Fig. 1g. Steel-Dwass test was used in Figs 2c, 3f, Supplementary Figs. 1a, 2g, and 3g. Relative risk analysis compared with control was used in Fig. 2f. Kruskal-Wallis test with Bonferroni correction was used in Fig. 3g. Wilcoxon signed rank test was used in Figs. 3h, i, 7e, and Supplementary Fig. 1d, with multiple comparisons with Bonferroni correction used in Fig. 5d, g, j, and Supplementary Fig. 5d. Significance was noted as ***$p < 0.001$, **$p < 0.01$, and *$p < 0.05$, and non-significant values were noted as ns. R version 3.5.0 was used for all non-parametric statistical analyses in this study[52].

## Data availability
We claim all relevant data are available from the authors with reasonable request.

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

# ARTICLE

49. Franklin, K. B. J. & Paxinos, G. *The Mouse Brain in Stereotaxic Coordinates.* (Academic Press, Cambridge, Massachusetts, USA 2007).
50. Osakada, F. & Callaway, E. M. Design and generation of recombinant rabies virus vectors. *Nat. Protoc.* **8**, 1583–1601 (2013).
51. Guenthner, C. J., Miyamichi, K., Yang, H. H., Heller, H. C. & Luo, L. Permanent genetic access to transiently active neurons via TRAP: targeted recombination in active populations. *Neuron* **78**, 773–784 (2013).
52. R. Core Team. *R: A Language and Environment for Statistical Computing.* (R Foundation for Statistical Computing, Vienna, 2018. https://www.R-project.org/

## Acknowledgements

We thank L. Luo for sharing *CAV2-FLEx-Flp*, and *AAV-hSyp-FLEx^{FRT}-mGFP*, E. Callaway for sharing B7GG, BHK-EnvA and HEK293-TVA800 cell lines, M. Yamaguchi for sharing rabies virus, W. Fujii for helps in designing gRNAs for CRISPR genome editing, RIKEN BSI Research Resource Center for help in generating knockout mice, and members of the Touhara lab for their helps. T.O. and K.K.I. are supported by research fellowship for young scientist from JSPS. This work was supported by ERATO Touhara Chemosensory Signal Project to K.T. (JST, grant number JPMJER1202), JSPS Kakenhi grant number 16K20963 and 17H05552 to K.M, and NIH funding RO1 DC013289 to S.D.L.

## Author contributions

T.O., K.K.I., K.M. and K.T. designed the study. T.O. identified the ESP22 receptor, generated mutant mice with a help from R.E. and Y.Y., and together with H.M. performed sexual behavior assays. K.M. performed viral injections and most histochemical analyses in the brain. K.K.I. performed optogenetic experiments. H.M. also performed reproduction assay. D.M.F. and S.D.L. provided some preliminary data. T.O., K.K.I., K.M. and K.T. wrote the paper with substantial contributions from other authors.

## Additional information

**Competing interests:** The authors declare no competing interests.

