## [Peer Review File · Nature Communications]

Reviewers' comments:

Reviewer #1 (Remarks to the Author):

The manuscript from Dr. Touhara and colleagues reports that the previously identified juvenile pheromone, ESP22, inhibits female mouse lordosis. Previously, it was shown to inhibit male sexual behavior by activation of vomeronasal sensory neurons. Here they identify the receptor, V2Rp4 that mediates the suppression of female lordosis, as knockout of the receptor eliminates the effect of ESP22. Further, they investigate the downstream circuitry involved in the behavior by chemogenetic loss of function experiments, and find that the medial amygdala and BNST are required for the ESP22 modulation of behavior. The connection from the BNST to the hypothalamus is via a population of GABAergic neurons that project to the VMHvl, which appears to inhibit sexual behavior, but I have some concern over this conclusion. Overall, this is an interesting paper and well-done set of experiments. I have the following comments and concerns.

1. In the text (Pg 4, line 13-15), the authors suggest "the main target of ESP22 secreted by 2 to 3 week old juveniles mice is not their own mothers, but rather other non-breeding female mice..." It is unclear how they reach that conclusion. The fact that juveniles can affect non-breeding females, does not mean that this is the main target. The authors should clarify or temper this conclusion.
2. Figure 2f: The authors show that ESP22 exposure leads to females with decreased pregnancy and increased delay to birth of pups. It is a little unclear how much of these changes are due to a lack of mating or decreased probability of conception. It would be helpful if the authors could clarify this issue.
3. Figure 5e, h: The statistics are a bit unclear. First the authors state that there is a significant negative correlation in the rejection ratio vs hM4Di expression level. How do they determine "significant negative correlation?" Second, the T-tests performed in Fig 5d,g,e only compares the CNO group control vs ESP22. We should also know if there is a significant difference in the CNO group vs saline group. This seems important given the variability in the data in Fig 5d, which makes this result difficult to believe.
4. Figure 7: The authors express ChR2 in GABAergic neurons of the BNST and stimulate their cell bodies or their terminals in VMHvl. They nicely confirm and quantify activation at the cell bodies in the BNST using c-fos expression, but they do not provide any evidence that the terminal stimulation works. Moreover, given that axon terminal stimulation can notoriously generate back propagating action potentials, the authors should determine if their stimulation activates the terminals in VMHvl or also the BNST cell bodies. If stimulation of the terminals activates the entire cell, they can only conclude that BNST neurons projection to VMHvl are sufficient to suppress lordosis behavior, rather than this projection is sufficient. Alternatively, this conclusion could be strengthened by demonstrating that activation of GABAergic BNST neurons, suppresses activity in the VMHvl, which is correlated with the suppression of behavior.
5. The model in Figure 7f: In the legend they should reference the previous work for the red arrows or clarify that this conclusion is based on previous work.

Reviewer #2 (Remarks to the Author):

This manuscript by Osakada and colleagues extensively documents the role of the juvenile hormone

ESP22 in inhibiting female sexual behavior. The authors demonstrate that exposure to ESP22 decreases female sexual behavior, as assessed by quantification of lordosis and active rejection actions. They then describe the identification of V2Rp4 as the receptor for ESP22, generate knockout lines for V2Rp4 and V2Rp6 via CRISPR, and show decreased rejection in females mutant for V2Rp4. The authors go on to map ESP22-responsive circuitry and demonstrate that activation of inhibitory BNST neurons, and their projections to the VMHvl results in increased rejection behavior, which they interpret as sexual suppression. Overall this is a very impressive and compelling manuscript that will be of significant interest to the field. However, I have a few issues with the specificity of the stereotactic injections, and the interpretation of the resulting data.

The retrobead experiments in the VMHvl demonstrate substantial inputs to the VMH from both the posterior (BNSTMP) and anterior (BNSTMA) BNST (Fig6k,l). However, many of the injections (Fig 5b, 6b,c, 7a,b) and all the fiber locations of ChR2-injected animals in Supp Fig 6, are targeted exclusively to the BNSTMA. This region is functionally distinct from the BNSTMP and is primarily interconnected with the central amygdala and the MeAad, not the MeApd/MeApv which project to the VMHvl and VMHdm, respectively (Dong and Swanson 2006 J Comp Neuro 494:142-178; Dong et al 2001 Brain Res Brain Res Rev; Canteras et al 1995 J Comp Neuro; Choi et al 2005 Neuron). However, the DREADD inhibition targeting data in Figure S4 suggest that both the MA and MP are involved in rejection behavior. Furthermore, the model presented by the authors in Fig 7 does not distinguish between these two BNST subdivisions. Can the authors please clarify in the text as to their interpretation of the role of ESP22 and the BNSTMA in the MeApd ->BNSTMP->VMHvl and the MeApv->VMHdm pathways?

The authors use inhibition with DREADDs to test the hypothesis that activity in distinct brain regions is essential for ESP22 suppression of sexual receptivity. Using a similar strategy, the authors previously demonstrated that inhibiting the MeA blocked the lordosis-enhancing properties of ESP1 (Ishii 2017 Neuron). However, in the present study, the DREADD data are inconsistent. In panels 5c and 5i, even control animals given saline show very low receptivity, as demonstrated by the low number of lordosis events and moderate number of active rejections. This is compared to raster plots for control animals in Figures 1b and 2c. Accordingly, the BNST data are more convincing than those from the MeA, as assessed by a significant drop in rejections between saline and CNO animals (Fig. 5g). The authors state that of the three MeA subdivisions: the MeApd, MeApv, and MeAa, only injections targeting the MeApv showed a significant correlation with behavior (Fig. S4). I am surprised that this is a sufficient number of hits to call significance as the *total* number of animals receiving CNO after MeA injections is 9 (Fig 5 legend). How many of these 9 animals have injections targeted to the MeApv?

Finally, it is clear from the c-fos experiments in Figure 4 that only a subset of the MeApv, the BNST, and the VMHdm are activated by ESP22. However, the expression of hM4Di is quite broad within these regions, therefore treatment with CNO inactivates most of the region-of-interest, not just the ESP22-responsive neurons. This would presumably interfere with the processing of all pheromonal cues such as ESP1, not just ESP22. As the authors note "considerable variabilities" in their data, I suggest that these experiments may not need to be included in the manuscript, and that the tracing and optogenetic data are sufficient.

Reviewer #3 (Remarks to the Author):

In the study of Osakada et. al., the authors found a novel function of the ESP22, a peptide pheromone secreted by juvenile mice. They found that ESP22 can suppress sexual receptivity of the virgin females and identified the key VNO receptor for ESP22. Immediate early gene studies revealed that ESP22

activated MEApv, BNST and VMHd. Functional manipulation demonstrated the necessity of BNST and MEApv but not VMHd in the behavioral effect of ESP22. In particular, the inhibitory BNST projection to the VMHvl, a key region for female sexual behaviors, is sufficient to mimic the effect of ESP22. The paper is interesting and novel. The reviewer only has several minor comments regarding the behaviors.

1. The status of the female. It is not very clear whether the females used in the study is OVX females with hormone treatment or naturally cycling females with hormone treatment. Given that the female sexual behavior is significantly modulated by the estrous cycle, it is essential that all the females have the same estrous status during testing.

2. In all raster plot figures (e.g. Figure 1b), only blue and red ticks are labeled but not gray ones. Please label.

3. The authors should provide some additional references regarding the categorization of the female sexual behaviors especially the definition of "rejection". For example, why standing and crouching are considered as rejection, but running away, kicking and squeaking are not? It will be helpful to have multiple experimenters to annotate the same videos to understand the agreement between human annotators. It will be also helpful to examine the rejection and lordosis behaviors across the estrous cycle to see whether the rejection is a useful parameter (like lordosis) to assess the receptive status of the animals. For example, does the animal show higher rate of rejection during diestrus than metestrus?

Reviewer #4 (Remarks to the Author):

The manuscript "Sexual rejection via a vomeronasal receptor-triggered limbic circuit" by Osakada et al. provides evidence of a circuit underlying pheromone-mediated sexual rejection in mice. Through a series of well executed experiments, the authors demonstrate that the peptide ESP22 secreted by juvenile mice is detected by the receptor V2Rp4 in the vomeronasal organ of adult females, and leads to suppression of reproductive behavior through a circuit involving MeA, BNST and VMHvl.

This provides evidence for a mechanism by which sexual reproduction can be suppressed if resources are shared amongst many and the conditions to foster offspring are not ideal. This study is one of the few analyzing the neural basis for sexual rejection, and the first to identify the limbic circuit subserving this function for ESP22 mediated-rejection. The work is technically sound and the results are convincing.

There are a few issues with the manuscript as it stands:

1) Given the importance that this circuit seems to play in mediating sexual rejections of female mice, why are the main behavioral experiments conducted only with virgin females? One would predict that this mechanism should serve its function in every female of the group, and therefore it would be helpful to see this behavior conserved in experienced females as well.

2) Even though the results are convincing, the statistics used to assess their significance are not always correct given the data. In most cases the authors use Student's t tests. However, these can only be used if the distribution of the data is normal or there are at least 30 data points per group to be compared – please test for Gaussianity. This doesn't seem to be the case for many of the comparisons. A non-parametric test, such as a Wilcoxon signed-rank test, should be used in those

situations instead. Furthermore, the use of a repeated measures ANOVA in Fig. 2c is incorrect, given that the sexual behavior of the females is not quantified across different days (and if it is, this should be clarified). Finally these are multiple draws from the exact same video – multiple comparisons corrections should be used throughout.

3) The conclusions in lines 13-15 of p.4 corresponding to the results of the experiment in Sup. Fig. 1e-g go beyond what the results of the experiment show. Lactating rodent mothers usually exhibit aggressive behaviors towards males. Some studies have shown this to be mediated by oxytocin and vasopressin (<https://www.ncbi.nlm.nih.gov/pmc/articles/PMC3826214/>), and this might be behind the sexual rejection. However, it is possible that ESP22 secreted by a mother's own pups may serve as a reinforcement or redundant mechanism to ensure reproductive suppression.

Minor issues

4) Given that the experiment shown in Sup. Fig. 3 c-e reveals that neurons projecting from MeApv to BNST are active upon stimulation with ESP22 and that this is a key part of the model schematized in Fig. 7f, maybe it should be moved to a main figure.

5) The legend of Sup. Fig. 4b should explain the meaning of the correlations, since it is crucial to understand the figure.

6) Given that the model shown in Fig. 7f brings together all of the results found in this study, the panel should be bigger. The space can be gained by making the raster plots in Fig. 7d smaller.

7) In the denominator of the index used for catFISH in Sup. Fig. 3f and g, why doesn't the denominator have the total number of cells? (ie, also add ESP1 alone).

8) Sup. fig. 1 needs statistical details in the legends. The methods state that "non-significant values were noted as n.s." but they are not. Modify accordingly.

9) Please add the strain of virgin females used in the legend of Fig. 1 and in the methods section.

10) The two bar plots in Fig. 7c are not necessary. Showing one should be enough.

11) In line 22 of p. 11, at the end of the line, there is an extra space between the word "described" and the citation.

12) In line 27 of page 9, it should say "...provide key insights on how information...".

13) The title of Sup. fig.1 shouldn't say "virgin female mice" given that the experiment shown in panels e-g was done with mothers.

We greatly thank all reviewers for the enthusiasm about our study, and for their constructive comments. We have performed additional experiments and data analyses (see below) in response to the comments, which have resulted in significant improvement of the paper. Below we first summarize the major changes in this revision, and then provide point-by-point responses to the reviewer's comments.

Additional experiments performed:

- 1) New optogenetic experiments to test if axon terminal stimulation of *GAD2+* BNST neurons in the VMHvl generates back-propagating activation of BNST cell bodies, as described in new **Supplementary Fig. 7**.
- 2) Additional optogenetic and histochemical experiments to test if activation of *GAD2+* BNST neurons could suppress *c-Fos+* expression in the VMHvl of female mice during mating, as discussed in the **Revise Fig. 2**.
- 3) Characterizing rejection behavior of female mice in estrus and diestrus states, as shown in **Revise Fig. 4**.

Additional analyses performed:

- 1) New statistical analyses in Figs. 1c, f, 2c, 3f, g-i, 4b, d, f, 5d, g, j, 6l, 7c, e, Supplementary Fig. 1a, b, d, g, 2g, 3b, e, g, 5d, and 7d as shown in **Table 1**.

Table 1 Non-parametric statistical analyses used in the revised manuscript.

Figure #	Statistical analysis
1c	Wilcoxon rank sum test with Holm correction
1f	Wilcoxon rank sum test with Holm correction
2c	Steel-Dwass test
3f	Steel-Dwass test
3g	Kruskal-Wallis test with Bonferroni correction
3h, i	Wilcoxon signed rank test
4b, d, f	Wilcoxon rank sum test
5d, g, j	Wilcoxon signed rank test with Bonferroni correction
6l	Wilcoxon rank sum test
7c	Wilcoxon rank sum test
7e	Wilcoxon signed rank test
S1a	Steel-Dwass test
S1b	Wilcoxon rank sum test
S1d	Wilcoxon signed rank test
S1g	Wilcoxon rank sum test with Holm correction
S2g	Steel-Dwass test
S3b,e	Wilcoxon rank sum test
S3g	Steel-Dwass test
S5d	Wilcoxon signed rank test with Bonferroni correction
S7d	Wilcoxon rank sum test

- 2) Testing consistency of rejection ratio among human annotators, as discussed in **Revise Fig. 3**.

Point-by-point responses:

Reviewer #1 (Remarks to the Author):

The manuscript from Dr. Touhara and colleagues reports that the previously identified juvenile pheromone, ESP22, inhibits female mouse lordosis. Previously, it was shown to inhibit male sexual behavior by activation of vomeronasal sensory neurons. Here they identify the receptor, V2Rp4 that mediates the suppression of female lordosis, as knockout of the receptor eliminates the effect of ESP22. Further, they investigate the downstream circuitry involved in the behavior by chemogenetic loss of function experiments, and find that the medial amygdala and BNST are required for the ESP22 modulation of behavior. The connection from the BNST to the hypothalamus is via a population of GABAergic neurons that project to the VMHvl, which appears to inhibit sexual behavior, but I have some concern over this conclusion. Overall, this is an interesting paper and well-done set of experiments. I have the following comments and concerns.

1. In the text (Pg 4, line 13-15), the authors suggest “the main target of ESP22 secreted by 2 to 3 week old juveniles mice is not their own mothers, but rather other non-breeding female mice...” It is unclear how they reach that conclusion. The fact that juveniles can affect non-breeding females, does not mean that this is the main target. The authors should clarify or temper this conclusion.

We agree with this opinion. According to the suggestion, we replaced the sentence with “one of the targets of ESP22 secreted by 2- to 3-week-old juvenile mice is non-breeding female mice in their environment” (Page 4, line 15-17).

2. Figure 2f: The authors show that ESP22 exposure leads to females with decreased pregnancy and increased delay to birth of pups. It is a little unclear how much of these changes are due to a lack of mating or decreased probability of conception. It would be helpful if the authors could clarify this issue.

Based on behavioral data described previously (Ferrero *et al.*, *Nature* 2013)¹ and in **Fig. 1** of this manuscript, ESP22 exposure suppresses both mounting of male mice and receptivity of female mice, which we speculate the main cause of decreased pregnancy and delayed birth of pups observed in **Fig. 2e-f**. However, we could not exclude the possibility of other uncharacterized effect(s) of ESP22 (e.g., decreased probability of conception) that may also contribute to delay to birth of pups. Whether or not ESP22 impacts conception probability, it seems reasonable to conjecture that ESP22-mediated inhibition of male and female sexual behavior contributes to decreased pregnancy and delayed birth, at least in part. To make our interpretation clear, in the revised manuscript, we added words

“presumably via decreased mounting of male mice¹ and receptivity of female mice (Fig. 1),” in our conclusion sentence of **Fig. 2** (Page 4, line 34-35).

3. Figure 5e, h: The statistics are a bit unclear. First the authors state that there is a significant negative correlation in the rejection ratio vs hM4Di expression level. How do they determine “significant negative correlation?” Second, the T-tests performed in Fig 5d,g,e only compares the CNO group control vs ESP22. We should also know if there is a significant difference in the CNO group vs saline group. This seems important given the variability in the data in Fig 5d, which makes this result difficult to believe.

We thank the reviewer for these comments. Negative correlation means that correlation coefficient (R) is negative and “significance” is determined by t-test on the null hypothesis that R is indistinguishable from zero. To make these points clear, we added a word “statistically” and the p value ($p = 0.03$) in the revised manuscript (Page 7, line 23-24).

Second, comparing ESP22-stimulated CNO group with ESP22-stimulated saline group is important to interpret the result of this loss-of-function experiment. Indeed, in the original manuscript, we showed a significant difference in the rejection ratio of ESP22-CNO vs ESP22-saline group in the loss-of-function of BNST neurons (**Fig. 5g**). In the MeA loss-of-function experiments (**Fig. 5c-e**), however, due to the large variability of in the rejection ratio of each trials (not only ESP22 exposure but also control exposure groups), no significant difference was found in the rejection ratio between the ESP22-CNO and ESP22-saline groups. In the original manuscript, we inferred this situation by saying “Although we observed considerable variabilities in rejection ratio in some of sexual behavior assays using these mice, presumably due to viral injection surgery-associated stress, control groups in which saline was intraperitoneally injected before the assay still showed ESP22-induced significant increase of rejection ratio.” Intra-group comparison (e.g., ESP22-CNO vs. control-CNO) is based on the same animals, and therefore less suffered from individual variations, which is more problematic in the case of inter-group comparison (e.g., ESP22-CNO vs. ESP22-saline). Therefore, we think that it is still reasonable to suggest that loss-of-function of MeA neurons tends to suppress ESP22-mediated sexual rejection, under an appropriate note about unavoidable variations of female sexual behaviors in our experimental condition. In the revised manuscript, we added statistical analyses of ESP22-CNO vs. ESP22-saline groups in MeA and VMH experiments (**Fig 5d and 5j**).

4. Figure 7: The authors express ChR2 in GABAergic neurons of the BNST and stimulate their cell bodies or their terminals in VMHvl. They nicely confirm and quantify activation at the cell bodies in the BNST using c-fos expression, but they do not provide any evidence that the terminal stimulation works. Moreover, given that axon terminal stimulation can notoriously generate back propagating action potentials, the authors should determine if their stimulation activates the terminals in VMHvl or also the BNST cell bodies. If stimulation of the terminals activates the entire cell, they can only conclude that BNST neurons projection to VMHvl are sufficient to suppress lordosis behavior, rather than this projection is sufficient. Alternatively,

this conclusion could be strengthened by demonstrating that activation of GABAergic BNST neurons, suppresses activity in the VMHvl, which is correlated with the suppression of behavior.

We thank the reviewer for this comment which is critical to interpret our optogenetic experiments. The reviewer has pointed out that there is no evidence that terminal stimulation of ChR2+ axons at VMHvl, 1) does not generate back propagating action potentials to BNST, and 2) inhibits post-synaptic neurons in the VMHvl.

In the revised manuscript, we conducted new cohorts of optogenetic experiments to answer point 1) above. We injected a Cre-dependent AAV driving either ChR2 or GFP into the BNST of *GAD2-Cre* male mice (**Revise Fig. 1A**). An optic-fiber was placed above VMHvl following viral injection. The locations of fibers were post-hoc analyzed and mapped on the brain atlas (Franklin and Paxinos, 2007)² as shown in **Revise Fig. 1B**. 2-weeks after viral injection, animals received 25 sec x 5 of light stimulation (20 ms, 20Hz, 3mW), which is identical to the protocol used in the experiments shown in **Fig. 7**. Thirty-minutes after light stimulation, animals were sacrificed to collect brain tissue. We found no light induced increase in *c-Fos* expression in the BNST of these animals (cohort 1 of **Revise Fig. 1C, D**). In contrast, positive control experiments in which *GAD2+* BNST neurons were targeted to drive ChR2 and light-stimulated via optic-fibers placed above BNST showed significant increase in *c-Fos* expression in the BNST (cohort 2), confirming our original observation in **Fig. 7c**. These results suggest that light stimulation of *GAD2+* BNST axon fibers in the VMHvl does not significantly activate BNST cell bodies by back-propagating action potentials.

Regarding point 2), we fully agree that direct demonstration of VMHvl inhibition by BNST GABAergic input would greatly strengthen the manuscript. This, however, requires patch-clamp recording of defined post-synaptic neurons in the brain slice upon light stimulation, which is beyond the scope of this manuscript and our technical expertise. Nevertheless, during the revision, we made efforts to collect supporting evidences of VMHvl inhibition. We bilaterally targeted ChR2 to *GAD2+* BNST neurons and light stimulated their axons in the VMHvl when the female mice were exposed with male mice (which is known to activate VMHvl neurons, see Nomoto & Lima, *Curr. Biol.* 25, 589, 2015)³. We, however, did not observe inhibition of *c-Fos* expression in the light stimulated animals (**Revise Fig. 2**). This failure may be due to the heterogeneity of VMHvl neurons (Remedios *et al.*, *Nature* 2017; Nomoto & Lima, *Curr. Biol.* 2015)^{3,4}: both receptive and rejective behavioral episodes during mating activate VMHvl neurons. Therefore, BNST GABAergic input to VMHvl may not always suppress global activities of VMHvl evoked by male mice.

Taken together, we think that it is reasonable to suggest that our observed sexual rejection is mediated by inhibition of VMHvl neurons, because terminal stimulation robustly affected female's sexual receptivity (**Fig. 7e**), and this effect was not due to global activations of BNST cell bodies (**Revise Fig. 1C, D**). However, due to the lack of direct evidence of this inhibition, we decided to tone down in the conclusion sentence of the manuscript by deleting words "inhibits VMHvl", and instead said "ESP22 recruits the inhibitory BNST to VMHvl pathway". We made new **Supplementary Fig. 7** to report the results discussed in **Revise Fig. 1**.

[Figure Redacted]

[Figure Redacted]

5. The model in Figure 7f: In the legend they should reference the previous work for the red arrows or clarify that this conclusion is based on previous work.

We added the reference (Ishii *et al.*, *Neuron* 2017)⁵ of our previous work about ESP1 in the legend of **Fig. 7f**.

Reviewer #2 (Remarks to the Author):

This manuscript by Osakada and colleagues extensively documents the role of the juvenile hormone ESP22 in inhibiting female sexual behavior. The authors demonstrate that exposure to ESP22 decreases female sexual behavior, as assessed by quantification of lordosis and active rejection actions. They then describe the identification of V2Rp4 as the receptor for ESP22, generate knockout lines for V2Rp4 and V2Rp6 via CRISPR, and show decreased rejection in females mutant for V2Rp4. The authors go on to map ESP22-responsive circuitry and demonstrate that activation of inhibitory BNST neurons, and their projections to the VMHvl results in increased rejection behavior, which they interpret as sexual suppression. Overall this is a very impressive and compelling manuscript that will be of significant interest to the field. However, I have a few issues with the specificity of the stereotactic injections, and the interpretation of the resulting data.

The retrobead experiments in the VMHvl demonstrate substantial inputs to the VMH from both the posterior (BNSTMP) and anterior (BNSTMA) BNST (Fig6k,l). However, many of the injections (Fig 5b, 6b,c, 7a,b) and all the fiber locations of Chr2-injected animals in Supp Fig 6, are targeted exclusively to the BNSTMA. This region is functionally distinct from the BNSTMP and is primarily interconnected with the central amygdala and the MeAad, not the MeApd/MeApv which project to the VMHvl and VMHdm, respectively (Dong and Swanson 2006 *J Comp Neuro* 494:142-178; Dong et al 2001 *Brain Res Brain Res Rev*; Canteras et al 1995 *J Comp Neuro*; Choi et al 2005 *Neuron*). However, the DREADD inhibition targeting data in Figure S4 suggest that both the MA and MP are involved in rejection behavior. Furthermore, the model presented by the authors in Fig 7 does not distinguish between these two BNST subdivisions. Can the authors please clarify in the text as to their interpretation of the role of ESP22 and the BNSTMA in the MeApd ->BNSTMP->VMHvl and the MeApv->VMHdm pathways?

We greatly thank the reviewer for this insightful comment. A classical view as it was shown by the reviewer emphasized different connectivity of BNST MA and MP to their downstream targets. Our anatomical and *c-Fos* data (**Fig. 4, Fig. 6g, i**), however, suggested that both MA and MP were activated by ESP22, and contained the ESP22-responding neurons that project to VMHvl. Therefore, ESP22 seems to recruit multiple BNST subdivisions to provide a negative tone to VMHvl. To clarify this point, we modified the second paragraph of our discussion (Page 10, line 35 to Page 11, line 4) about the subdivisions of BNST.

As loss of function of VMHdm had no effect on ESP22-mediated rejection (**Fig. 5** and **Supplementary Fig. 5**), the MeApv to VMHdm pathway seems dispensable with processing ESP22. We do not exclude the possibility that direct projections from MeAa and MeApd to VMHvl play some roles in mediating ESP22 signals, in addition to multi-synaptic MeApv-BNST-VMHvl pathway. As we did not intend to test potential differences in the function of BNSTMA and MP in processing pheromones (which would be interesting topics for future studies), we did not distinguish BNST subdivisions in the model shown in **Fig. 7f**.

The authors use inhibition with DREADDs to test the hypothesis that activity in distinct brain regions is essential for ESP22 suppression of sexual receptivity. Using a similar strategy, the authors previously demonstrated that inhibiting the MeA blocked the lordosis-enhancing properties of ESP1 (Ishii 2017 Neuron). However, in the present study, the DREADD data are inconsistent. In panels 5c and 5i, even control animals given saline show very low receptivity, as demonstrated by the low number of lordosis events and moderate number of active rejections. This is compared to raster plots for control animals in Figures 1b and 2c. Accordingly, the BNST data are more convincing than those from the MeA, as assessed by a significant drop in rejections between saline and CNO animals (Fig. 5g). The authors state that of the three MeA subdivisions: the MeApd, MeApv, and MeAa, only injections targeting the MeApv showed a significant correlation with behavior (Fig. S4). I am surprised that this is a sufficient number of hits to call significance as the *total* number of animals receiving CNO after MeA injections is 9 (Fig 5 legend). How many of these 9 animals have injections targeted to the MeApv?

We sincerely apologize that the figure legend of our original **Supplementary Fig. 4** was not clear about the data analysis we conducted. Our loss-of-function experiments shown in **Fig. 5** and **Supplementary Fig. 4** were based on correlation analysis, and not on specific targeting of sub-divisions. For example in the MeA, we injected DREADD-Gi virus to target MeApv, but in most of the virus injections, the virus spread to all sub-regions of the MeA (a, pd and pv) with certain variability. We then analyzed correlation coefficient (R) between the ratio of mCherry+ pixel (inferring hM4Di targeting efficiency) in each sub-region and Δ rejection ratio (showing the effect of ESP22). This allowed us to show in **Fig. 5e** and **Supplementary Fig. 4b** a trend that the more mCherry (hM4Di) expression in MeApv, the more intensively ESP22-mediated sexual rejection was suppressed. To clarify these points, in the revised manuscript, we added detailed explanation of our method in the legend of **Supplementary Fig. 4**.

Regarding the variability of female's sexual behaviors in our viral loss-of-function experiments (**Fig. 5**) compared with wild type cases (**Fig. 1**), we were aware of the concerns pointed out by the reviewer. In the original manuscript, we mentioned in the explanation of **Fig. 5** that "we observed considerable variabilities in rejection ratio in some of sexual behavior assays using these mice, presumably due to viral injection surgery-associated stress". Intra-group comparison (e.g., ESP22-CNO vs. control-CNO) is based on the same animals, and therefore less suffered from individual variations, while inter-group comparison (e.g., ESP22-CNO vs. ESP22-saline) seems more

problematic. Therefore, we think that it is still reasonable to suggest that loss-of-function of MeA neurons tends to suppress ESP22-mediated sexual rejection based on intra-group comparison, under an appropriate note about unavoidable variations of female sexual behaviors in our experimental condition.

Finally, it is clear from the c-fos experiments in Figure 4 that only a subset of the MeApv, the BNST, and the VMHdm are activated by ESP22. However, the expression of hM4Di is quite broad within these regions, therefore treatment with CNO inactivates most of the region-of-interest, not just the ESP22-responsive neurons. This would presumably interfere with the processing of all pheromonal cues such as ESP1, not just ESP22. As the authors note “considerable variabilities” in their data, I suggest that these experiments may not need to be included in the manuscript, and that the tracing and optogenetic data are sufficient.

We fully agree with this reviewer that our manipulation is not specific to ESP22-responsive neurons and restricting loss-of-function experiments in functionally defined sub-population would greatly improve the resolution of circuit dissection. However, manipulating specific sub-population based on activity patterns itself is a quite challenging topic of neuroscience. Indeed, most of our knowledge of brain functions are based on crude loss-of-function experiments (lesion, drug injection, etc.) and studies of postmortal human brains. Most of current pharmaco- and opto-genetic loss-of-function experiments in mice are conducted at the level of brain region or a cell type defined by expression of Cre, but not exclusively on functionally defined sub-population. Since testing necessity of brain region/cell types is an important step forward to understanding how brain processes a specific sensory input, we believe that representing our crude loss-of-function data provides some useful insights into function of MeA-BNST-VMHvl axis in mediating ESP22 signals.

Reviewer #3 (Remarks to the Author):

In the study of Osakada et. al., the authors found a novel function of the ESP22, a peptide pheromone secreted by juvenile mice. They found that ESP22 can suppress sexual receptivity of the virgin females and identified the key VNO receptor for ESP22. Immediate early gene studies revealed that ESP22 activated MEApv, BNST and VMHd. Functional manipulation demonstrated the necessity of BNST and MEApv but not VMHd in the behavioral effect of ESP22. In particular, the inhibitory BNST projection to the VMHvl, a key region for female sexual behaviors, is sufficient to mimic the effect of ESP22. The paper is interesting and novel. The reviewer only has several minor comments regarding the behaviors.

1. The status of the female. It is not very clear whether the females used in the study is OVX females with hormone treatment or naturally cycling females with hormone treatment. Given

that the female sexual behavior is significantly modulated by the estrous cycle, it is essential that all the females have the same estrous status during testing.

We apologize that our original explanation about status of sexual cycle of female mice was not clear. In early studies of our group (Haga *et al.*, *Nature* 2010; Ishii *et al.*, *Neuron* 2017)^{5,6}, we intended to use OVX female mice with hormone prime to establish pseudo-estrus state. However, during the study of this manuscript (behavioral data shown in **Fig. 1**), we noticed that our previous surgery to remove ovary was insufficient. As a result, the female mice used in **Fig. 1** data should be interpreted as naturally cycling female mice with additional hormonal treatments (0.1 ml estrogen 24 and 48 h before the assay and with 0.05 ml progesterone 4 h before the assay) to mimic pseudo-estrus state. We confirmed that they indeed showed natural sexual cycles (data now shown). Nevertheless, the behavioral data shown in **Fig. 1** (and our previously published sexual behavioral assays) indicated that our procedures permitted us to observe female sexual behaviors and potential impacts of pheromone treatment. Therefore, throughout the manuscript, we did not change the procedures to prepare female mice for sexual behavioral assay. To make this point clear, in the revised manuscript, we modified methods section as shown below.

[Page 14, line 7-14]

Female sexual behavior assay. (Figs. 1, 2a–c, 3g–i, 5, 7a–e, Supplementary Figs. 1e–g, and 5) Sexual behaviors of female mice were analyzed, as has been previously described^{5,6}. Briefly, female mice with natural sexual cycles were additionally primed with 0.1 ml estrogen (Wako Pure Chemical Industries, 0.4 mg/ml in corn oil) 24 and 48 h before the assay and with 0.05 ml progesterone (Wako Pure Chemical Industries, 10 mg/ml in corn oil) 4 h before the assay to mimic their phase of estrus during the assay. In addition, handling treatment was performed at least 10 days to keep their calmness. Female mice were isolated two days before the assay.

2. In all raster plot figures (e.g. Figure 1b), only blue and red ticks are labeled but not gray ones. Please label.

According to this suggestion, we added the information of gray ticks in all raster plot figures, “gray bars represent mount without lordosis or rejection”.

3. The authors should provide some additional references regarding the categorization of the female sexual behaviors especially the definition of “rejection”. For example, why standing and crouching are considered as rejection, but running away, kicking and squeaking are not? It will be helpful to have multiple experimenters to annotate the same videos to understand the agreement between human annotators. It will be also helpful to examine the rejection and lordosis behaviors across the estrous cycle to see whether the rejection is a useful parameter (like lordosis) to assess the receptive status of the animals. For example, does the animal show higher rate of rejection during diestrus than metestrus?

We thank the reviewer for this insightful comment. We sincerely apologize that our original explanation about rejection was insufficient for readers to understand our definition. We defined “rejection” if we observe at least one of the following postures of female mice during male mounting: standing, crouching, keeping their limb tight, or turning their body, to avoid male’s insertion, because these postures were frequently observed in the trials of ESP22 exposure. On the other hand, running away, kicking, and squeaking were not included in our definition of rejection because these postures were rarely elicited by ESP22. In the revised manuscript, we clarified this point by changing our main text (Page 3, line 30-33) and the corresponding method section (Page 14, line 25-26).

[Page 3, line 30-33]

Intriguingly, we observed that an ESP22-stimulated female mouse frequently showed at least one of the following postures: standing, crouching down, keeping their limb tight, or turning their body, to avoid insertion when male mice tried to mount onto female mice (Supplementary video 1). We defined this behavior as “rejection.”

Next, to test the agreement about definition of rejection, we analyzed the same video by multiple experimenters who were blinded regarding the experimental conditions. As shown below (**Revise Fig. 3**), overall, two human annotators reported similar numbers of rejection episodes, with certain variability in the number of male mount attempt as well. In the case of GAD2::GFP-BNST data (in which female mice were not rejective), annotator #1 and #2 reported rejection ratio of 15% and 28%, respectively. In the case of GAD2::Chr2-BNST sample (in which female mice were highly rejective), both annotators agreed with very high rejection ratio (94% and 95%). Thus, we think that certain variability is inevitable, in particular, when the rejection ratio is low. This may account for some basal variability of rejection ratio between wild type (**Fig. 1**) and pharmaco-genetically manipulated mice (**Fig. 5**). Throughout the manuscript, a single human annotator (who was blinded regarding the experimental condition) was assigned to analyze the data within the same figure, and therefore, variability across annotators did not significantly affect our data analysis and conclusions.

Revise Fig. 3 Analyzation same video of behavioral trials by multiple experimenters.

Raster plots by multiple experimenters (Analyzer #1 and #2) representing mounting episodes made by the male mouse, with cyan and yellow bars representing attempts associated with rejection or lordosis responses, respectively, by female mice. Mounting attempts without lordosis or rejection are represented with black bars.

Next, to address if our rejection can be a useful parameter to assess the receptivity of female mice, we performed a new cohort of behavior experiments using C57BL/6 female mice without any surgery or hormone priming (**Revise Fig. 4**). We performed sexual behavior assay as described in the method section, except daily handling and checking their estrus phase via vaginal smear from two days before the mating assay. We found that rejection ratio of female mice in their natural estrus phase was about 40% (**Revise Fig. 4B and C**), which is slightly higher, but within the range of rejection ratio we observed in pseudo-estrus phase (**Fig. 1**). In sharp contrast, females in their natural diestrus phase were highly rejective, as demonstrated by about 90% rejection ratio. Of note, we also observed very high rejection ratio in lactating mother (**Supplementary Fig. 1g**). Taken together, these results demonstrate that rejection defined in this study is a useful parameter to assess receptivity of female mice.

Revise Fig. 4 Rejection score of non-breeding female mice is dependent to their estrus phase.

A Timeline for sexual behavior assays using C57BL/6 virgin female mice without surgery or hormone priming. **B** Raster plots representing mounting episodes made by the male mouse, with cyan and magenta bars representing attempts associated with lordosis or rejection responses, respectively, by female mice. Mounting attempts without lordosis or rejection are represented with grey bars. Female mice in diestrus phase, $n = 3$; in estrus phase, $n = 3$. **C** Quantification of the sexual behaviors of female mice in the diestrus phase and in the estrus phase. Each dot represents data of an individual female mice. Error bars, S.E.M. $**p < 0.01$ by two-sided Student's t -test.

Reviewer #4 (Remarks to the Author):

The manuscript “Sexual rejection via a vomeronasal receptor-triggered limbic circuit” by Osakada et al. provides evidence of a circuit underlying pheromone-mediated sexual rejection in mice. Through a series of well executed experiments, the authors demonstrate that the peptide ESP22 secreted by juvenile mice is detected by the receptor V2Rp4 in the vomeronasal organ of adult females, and leads to suppression of reproductive behavior through a circuit

involving MeA, BNST and VMHvl. This provides evidence for a mechanism by which sexual reproduction can be suppressed if resources are shared amongst many and the conditions to foster offspring are not ideal. This study is one of the few analyzing the neural basis for sexual rejection, and the first to identify the limbic circuit subserving this function for ESP22 mediated-rejection. The work is technically sound and the results are convincing.

There are a few issues with the manuscript as it stands:

1) Given the importance that this circuit seems to play in mediating sexual rejections of female mice, why are the main behavioral experiments conducted only with virgin females? One would predict that this mechanism should serve its function in every female of the group, and therefore it would be helpful to see this behavior conserved in experienced females as well.

We thank the reviewer for this valuable comment. Robust sexual rejection was observed in experienced female mice when BNST GABAergic neurons (or their axon termini in the VMHvl) were opto-genetically activated. Thus, the function of BNST-VMHvl circuitry seems to be conserved in experienced females as well. To make this point clear, we added a few sentences at the end of Method section to note our observation (Page 15, line 15-18)

2) Even though the results are convincing, the statistics used to assess their significance are not always correct given the data. In most cases the authors use Student's t tests. However, these can only be used if the distribution of the data is normal or there are at least 30 data points per group to be compared – please test for Gaussianity. This doesn't seem to be the case for many of the comparisons. A non-parametric test, such as a Wilcoxon signed-rank test, should be used in those situations instead. Furthermore, the use of a repeated measures ANOVA in Fig. 2c is incorrect, given that the sexual behavior of the females is not quantified across different days (and if it is, this should be clarified). Finally these are multiple draws from the exact same video – multiple comparisons corrections should be used throughout.

We greatly appreciate the reviewer for this very important suggestion. We agree that the numbers of data points in our study were often too small to test Gaussianity. In the revised manuscript, we renewed our statistical analyses by using non-parametric test in most of our figure panels. Wilcoxon signed rank test was used in **Figs. 3h, i, 5d, g, j, 7e, Supplementary Figs. 1d, and 5d**. Non-repeated measures ANOVA previously used in **Fig. 2c** was changed to Steel-Dwass test, because as the reviewer pointed out, the female mice were not used repeatedly in **Fig. 2c**. We also applied corrections of multiple comparisons such as Holm correction in **Fig. 1c, f** and **Supplementary Fig. 1g**, because we have observed multiple elements (such as male mount and female rejection) from the same data set. We summarize all statistical changes in the revised manuscript in **Table 1**. These non-parametric statistical tests were conducted by using R version 3.5.0 (Ref. 7) as shown in the methods section (Page 21, line 7-8).

Overall, our original conclusions were supported by using non-parametric statistical tests. In small number of cases, however, the p values were slightly above 0.05 standard. In these cases (i.e., Figs. 1f, 3f, 5d, 6l and Supplementary Fig. 5d), we showed exact p values in the figure panel and toned down our conclusion states by adding “trend” in the main text.

Table 1 Non-parametric statistical analysis used in the revised manuscript.

Figure #	Statistical analysis
1c	Wilcoxon rank sum test with Holm correction
1f	Wilcoxon rank sum test with Holm correction
2c	Steel-Dwass test
3f	Steel-Dwass test
3g	Kruskal-Wallis test with Bonferroni correction
3h, i	Wilcoxon signed rank test
4b, d, f	Wilcoxon rank sum test
5d, g, j	Wilcoxon signed rank test with Bonferroni correction
6l	Wilcoxon rank sum test
7c	Wilcoxon rank sum test
7e	Wilcoxon signed rank test
S1a	Steel-Dwass test
S1b	Wilcoxon rank sum test
S1d	Wilcoxon signed rank test
S1g	Wilcoxon rank sum test with Holm correction
S2g	Steel-Dwass test
S3b,e	Wilcoxon rank sum test
S3g	Steel-Dwass test
S5d	Wilcoxon signed rank test with Bonferroni correction
S7d	Wilcoxon rank sum test

3) The conclusions in lines 13-15 of p.4 corresponding to the results of the experiment in Sup. Fig. 1e-g go beyond what the results of the experiment show. Lactating rodent mothers usually exhibit aggressive behaviors towards males. Some studies have shown this to be mediated by oxytocin and vasopressin (<https://www.ncbi.nlm.nih.gov/pmc/articles/PMC3826214/>), and this might be behind the sexual rejection. However, it is possible that ESP22 secreted by a mother’s own pups may serve as a reinforcement or redundant mechanism to ensure reproductive suppression.

We agree with this opinion. We replaced the conclusion sentence of **Supplementary Fig. 1** experience with “one of the targets of ESP22 secreted by 2- to 3-week-old juvenile mice is non-breeding female mice in their environment” (Page 4, line 15-17).

Minor issues

4) Given that the experiment shown in Sup. Fig. 3 c-e reveals that neurons projecting from MeApv to BNST are active upon stimulation with ESP22 and that this is a key part of the model schematized in Fig. 7f, maybe it should be moved to a main figure.

We greatly thank the reviewer for his/her interest in this data. We fully agree the importance of showing ESP22-responding neurons in the MeApv that send axonal projections to the BNST. However, we think that it is still fair to keep this data in the supplementary figure by two reasons. 1) We do not know if MeA to BNST projections really contribute to the activation of BNST by ESP22, because a direct AOB to BNST pathway exists (as inferred in **Fig. 7f**) and potentially many other indirect pathways can be considered. 2) ESP22 activates not only MeA neurons projecting to the BNST, but also those projecting to VMHd (although the latter did not reach to statistical significance). Compared with the case of ESP1 which selectively activates MeA neurons projecting VMHd (Ishii *et al.*, *Neuron* 2017)⁵, the pathway selectivity of ESP22 signal is not very clear.

5) The legend of Sup. Fig. 4b should explain the meaning of the correlations, since it is crucial to understand the figure.

We sincerely apologize that the figure legend of our original **Supplementary Fig. 4** was not clear about the data analysis we conducted. In the revised manuscript, we added detailed explanation of correlation analysis to the legend of **Supplementary Fig. 4** as shown below. (Updated text is shown by yellow highlight.)

Supplementary Fig. 4 Additional analysis of loss-of-function experiments. **a** Proportion of mCherry-positive pixels in each sub-region and whole MeA, BNST, and VMHd that were targeted in loss-of-function experiments. Error bars, S.E.M. As shown, mCherry (hM4Di) was broadly targeted to MeAa, MeApv and MeApd. In the BNST, mCherry was mainly targeted to MA, L, and MP subregions with minimum expression in MV or LV subregions. In the VMH, as SF1-Cre was used to restrict transgene expression only in the VMHd, almost all mCherry positive neurons were located in the VMHd. Each experimental animal showed certain variability in the targeting efficiency in each subregion, allowing us to analyze correlation coefficient (R) between ratio of mCherry+ pixel (inferring hM4Di targeting efficiency) and Δ rejection ratio (showing the effect of ESP22). Some of these analyses were graphically shown in Figures 5e, 5h, and 5k. **b** Table in this panel shows R values for each subregion and total MeA, BNST and VMH, with p-values (p) under the null-hypothesis that there is no correlation (R = 0). Red text represents $p < 0.05$. In the MeA, hM4Di-targeting efficiency in MeApv, but not MeApd, tended to negatively correlate with Δ rejection ratio, suggesting that loss-of-function of MeApv impaired ESP22-induced rejection. In the BNST, MA, L, and MP all showed trend of negative correlation, but R-value was only significantly different from zero when entire BNST data were pooled (total). In VMH, none of subregion showed negative correlation. Of note, we did not correct multiple comparisons in this table.

6) Given that the model shown in Fig. 7f brings together all of the results found in this study, the panel should be bigger. The space can be gained by making the raster plots in Fig. 7d smaller.

According to this suggestion, we changed the size of **Fig. 7f** much bigger than before.

7) In the denominator of the index used for catFISH in Sup. Fig. 3f and g, why doesn't the denominator have the total number of cells? (ie, also add ESP1 alone).

We thank the reviewer for this important question that we should have elaborated in the original manuscript. When the same stimulant was given twice (e.g., "ESP1-ESP1"), the nuclear transcripts positive cells (the second-stimulant responding cells) were reliably labeled with cytoplasmic mRNA (the first-stimulant responding cells), but many mRNA positive cells were not labeled with nuclear transcripts. Habituation to the same stimulation during the second application period, or less stable nature of induction of nuclear transcript may account for this observation. Whatever the mechanisms, the number of dual positive cells divided by the nuclear transcript positive cells (A-index) more robustly represents overlap in the neural representations, than the denominator having the total responding cells. Notably, previous studies by other groups also used the same index (Lin *et al.*, *Nature* 2011; Wu *et al.*, *Nature* 2014)^{8,9}.

We added this information in the method section of catFISH (Page 17, line 33 to Page 18, line 6).

8) Sup. fig. 1 needs statistical details in the legends. The methods state that "non-significant values were noted as n.s." but they are not. Modify accordingly.

The original supplementary Fig 1 data were not statistically analyzed. According to this suggestion, we applied non-parametric statistical analyses in **Supplementary Fig. 1a, b, d, and g**. We also added "ns" into the figure panels when we found no statistical significance.

9) Please add the strain of virgin females used in the legend of Fig. 1 and in the methods section.

We added the information of the strain of virgin females (C57BL/6J and C3H/HeJ) in the legend of **Fig. 1** and the methods section about behavioral experiments.

10) The two bar plots in Fig. 7c are not necessary. Showing one should be enough.

According to this suggestion, we only showed quantification of the number of *c-Fos* positive cells per section in the revised **Fig. 7c**.

11) In line 22 of p. 11, at the end of the line, there is an extra space between the word “described” and the citation.

12) In line 27 of page 9, it should say “...provide key insights on how information...”.

We thank the reviewer for finding out these errors. We have corrected them.

13) The title of Sup. fig.1 shouldn't say “virgin female mice” given that the experiment shown in panels e-g was done with mothers.

We have changed the title of **Supplementary Fig. 1**. “Effect of ESP22 on C57BL/6 virgin female mice and lactating mothers.”

References

1. Ferrero, D. M. *et al.* A juvenile mouse pheromone inhibits sexual behaviour through the vomeronasal system. *Nature* **502**, (2013).
2. Franklin, K. B. J. & Paxinos, G. *The mouse brain in stereotaxic coordinates*. (Academic Press, 2007).
3. Nomoto, K. & Lima, S. Q. Enhanced Male-Evoked Responses in the Ventromedial Hypothalamus of Sexually Receptive Female Mice. *Curr. Biol.* **25**, 589–594 (2015).
4. Remedios, R. *et al.* Social behaviour shapes hypothalamic neural ensemble representations of conspecific sex. *Nature* **550**, 388–392 (2017).
5. Ishii, K. K. *et al.* A Labeled-Line Neural Circuit for Pheromone-Mediated Sexual Behaviors in Mice. *Neuron* **95**, 123–127 (2017).
6. Haga, S. *et al.* The male mouse pheromone ESP1 enhances female sexual receptive behaviour through a specific vomeronasal receptor. *Nature* **466**, 118–22 (2010).
7. R Core Team. R: A language and environment for statistical computing. R Foundation for Statistical Computing, Vienna, Austria. <https://www.R-project.org/>. (2018).
8. Lin, D. *et al.* Functional identification of an aggression locus in the mouse hypothalamus. *Nature* **470**, 221–6 (2011).
9. Wu, Z., Autry, A. E., Bergan, J. F., Watabe-Uchida, M. & Dulac, C. G. Galanin neurons in the medial preoptic area govern parental behaviour. *Nature* **509**, 325–30 (2014).

REVIEWERS' COMMENTS:

Reviewer #1 (Remarks to the Author):

The authors have satisfied all of my concerns and the improved manuscript is now suitable for publication. The paper is well done and will make for a very nice paper.

Reviewer #2 (Remarks to the Author):

The authors have thoroughly responded to each of my comments by clarifying several points in the text, and elaborating on the legend in Supp Fig 4. I am satisfied with their explanations and I can now recommend this manuscript for publication.

Reviewer #3 (Remarks to the Author):

The authors have answered all my questions satisfactorily. I support publication of this study in Nature Communication.

Reviewer #4 (Remarks to the Author):

We congratulate the authors on a very nice revision - they have comprehensively dealt with all of our concerns, and the additional controls they have run on regarding the bAPs strengthens the manuscript considerably. This will be a nice contribution to the literature.